# Disaccharides and Fructooligosaccharides (FOS) Production by Wild Yeasts Isolated from Agave

**DOI:** 10.3390/foods14152714

**Published:** 2025-08-01

**Authors:** Yadira Belmonte-Izquierdo, Luis Francisco Salomé-Abarca, Mercedes G. López, Juan Carlos González-Hernández

**Affiliations:** 1Tecnológico Nacional de México/Instituto Tecnológico de Morelia, Av. Tecnológico #1500, Morelia 58120, Mexico; yadira.bi@morelia.tecnm.mx; 2Colegio de Postgraduados, Km. 36.5 Carretera Federal México-Texcoco Montecillo, Texcoco de Mora 56264, Mexico; luis.salome@colpos.mx; 3Centro de Investigación y de Estudios Avanzados del Instituto Politécnico Nacional, Km. 9.6 Carretera Irapuato-León, Irapuato 36824, Mexico

**Keywords:** fructooligosaccharides, blastose, 6-kestose, yeast

## Abstract

Fructooligosaccharides (FOS) are short fructans with different degrees of polymerization (DP) and bonds in their structure, generated by the distinct activities of fructosyltransferase enzymes, which produce distinct types of links. FOS are in high demand on the market, mainly because of their prebiotic effects. In recent years, depending on the link type in the FOS structure, prebiotic activity has been shown to be increased. Studies on β-fructanofuranosidases (Ffasa), enzymes with fructosyltransferase activity in yeasts, have reported the production of ^1^F-FOS, ^6^F-FOS, and ^6^G-FOS. The aims of this investigation were to evaluate the capability of fifteen different yeasts to grow in *Agave* sp. juices and to determine the potential of these juices as substrates for FOS production. Additionally, the research aimed to corroborate and analyze the fructosyltransferase activity of enzymatic extracts obtained from agave yeasts by distinct induction media and to identify the role and optimal parameters (time and sucrose and glucose concentrations) for FOS and disaccharides production through Box–Behnken designs. To carry out such a task, different techniques were employed: FT-IR, TLC, and HPAEC-PAD. We found two yeasts with fructosyltransferase activity, *P. kudriavzevii* ITMLB97 and *C. lusitaniae* ITMLB85. In addition, within the most relevant results, the production of the FOS 1-kestose, 6-kestose, and neokestose, as well as disaccharides inulobiose, levanobiose, and blastose, molecules with potential applications, was determined. Overall, FOS production requires suitable yeast species, which grow in a medium under optimal conditions, from which microbial enzymes with industrial potential can be obtained.

## 1. Introduction

Fructans are fructose polymers with linear or branched structures if any there is a D-glucose unit in the molecule [1,2]. Different types of fructans have been described and classified on the basis of their glycosidic linkage types: inulin (β(2→1)), levan (β(2→6)), graminans (β(2→1) and β(2→6)), neoseries of inulin and neoseries of levan (both with internal glucose) and agavins (β(2→1) and β(2→6)), highly branched fructans with internal glucose). In addition, fructans are classified on the basis of their degree of polymerization (DP) as fructooligosaccharides (FOS) if their DP is between DP3 and DP12 and as high-polymerization degree fructans (HDP-fructans) if they have a DP > 12. In particular, FOS are in greatest demand by the pharmaceutical and food industries because of their prebiotic activity and, therefore, their benefits to human health. For example, FOS can increase the absorption of minerals such as calcium, iron, and magnesium; increase beneficial intestinal microbiota and the inhibition of pathogens; decrease total cholesterol and serum lipids; positively stimulate the immune system; increase IgA secretion; decrease proinflammatory cytokines; and exert antioxidant effects [3,4,5,6,7]. FOS are produced by plants and microorganisms. In microorganisms, FOS synthesis is mediated by enzymes such as fructosyltransferases (Ftases, E.C. 2.4.1.9) and β-fructofuranosidases (Ffases, E.C. 3.2.1.26) [8]. The transfructosylation of fructose to sucrose or a FOS molecule results in an increase in the level of the molecule DP [9]. In wild yeasts, Ffase, also known as invertase, possesses fructosyltransferase activity; however, this activity requires high sucrose concentrations of approximately 3 M [10]. Microbial enzymatic activity results in a structural variety of FOS, including ^1^F-FOS (inulin-type FOS), ^6^F-FOS (levan-type FOS), ^1,6^F-FOS (graminan-type FOS), ^6^G-FOS (neo-levan FOS), and probably aFOS (agavin-FOS), which later polymerize into all types of HDP-fructans [11,12,13].

In this context, the most studied type of FOS is ^1^F-FOS, followed by ^6^F-FOS. However, little is known about the ^6^G-FOS [12,14]. When transfructosylation reactions occur, not only can sucrose act as an acceptor of the fructosyl group, but glucose or fructose can also do so. If glucose acts as an acceptor of the fructosyl group, a disaccharide called blastose is formed, which is considered the first construction block of neo-FOS and blasto-FOS, which, in addition, possesses potential in health [15,16,17]. This structural variation in FOS might be linked to distinctive effects on biological functions, as β(2→6) linkages enhance the prebiotic potential and chemical stability of these molecules compared with conventional ^1^F-FOS [12,18]. In addition, some reports of ^6^F-FOS include Ffase enzymes in *Saccharomyces cerevisiae* [19] and *Schwanniomyces occidentalis* [20]. Similarly, Ffase enzymes from *Phaffia rhodozyma* [21] and *Xanthophyllomyces dendrorhous* were found to produce ^6^G-FOS [22,23]. Interestingly, the ^6^G-Ffase of the yeast *X. dendrorhous* also produces blastose [13], which is a byproduct of FOS production. In this context, FOS production varies depending on the type of microorganism (bacteria, filamentous fungi, or yeast) and its species, as well as its enzymes. In addition, FOS production is influenced by parameters such as temperature, pH, agitation speed, nutrients, substrate, and culture medium composition [8,24,25].

On the other hand, fructans occur in approximately 15% of terrestrial plants. Some of the most well-known species include chicory, dahlia, onion, garlic, asparagus, and agave; the first photosynthetic products of agaves are fructans [26,27]. Agaves are emblematic species from Mexico, and approximately 77% of all agave species worldwide are in the country [28]. Nonetheless, only a few species, such as *Agave tequilana* var. azul, are industrially exploited to produce tequila. A wide variety of Agave species, including *A. salmiana*, *A. mapisaga*, *A. atrovirens*, *A. americana*, and *A. ferox*, are used to obtain other beverages, such as mezcal and pulque [29]. However, many other species and varieties have not yet been adequately explored or exploited, as their composition could be the basis for obtaining other products of interest, such as prebiotics, specifically FOS. In this context, only a few studies have used agave juice as a substrate for the evaluation of fructanase activity. The few ones include the hydrolytic activity of enzymes from *Kluyveromyces marxianus* and *Saccharomyces paradoxus* [30], as well as fructosyltransferase activity from *Aspergillus oryzae* [31]. However, agave juices could be explored as substrates to produce FOS through the fructosyltransferase activity of yeasts. The region of the lake of Cuitzeo in Michoacán, México, possesses peculiar features; for example, the highest degree of salinity of Mexico’s lakes is between 700 and 13,000 µS/cm [32,33,34]. This salinity affects the growth of most crops in this region, and agaves are not the exception. In addition, there are large wild agaves in the region; however, they are not used for any commercial purpose. Notably, the *Agave* sp. specimens in this region possess small oval-plane pine heads (Appendix A), but the variation in their fructans, especially FOS, has not been profiled. Additionally, the aguamiel (sap of the agave) is an important source of fructose, glucose, sucrose, fructans, amino acids, proteins, minerals, and vitamins [35]. Therefore, it could be interesting to explore the potential of agave juices from this region as a substrate to produce FOS by yeasts, which, until now, has remained unexplored. In addition, supported by experimental designs such as the Box–Behnken design, the systematic variation in various variables can be explored to find the optimal conditions for FOS production [36,37,38]. Therefore, this study aimed to evaluate the ability of fifteen different yeasts to grow in *Agave* sp. juices and to determine the potential of agave juices as a substrate for FOS production. Additionally, the research sought to corroborate and analyze the fructosyltransferase activity of enzymatic extracts obtained from agave yeasts via distinct induction media and to identify the role and optimal parameters (time, sucrose, and glucose concentrations) for FOS and disaccharides production through Box–Behnken designs.

## 2. Materials and Methods

### 2.1. Plant Material and Agave Juice Extraction

In different samples, a total of ten twelve-year-old wild *Agave* sp. specimens were collected in Cuitzeo, Michoacán, Mexico (19°57’45.5” N, 101°12’29.4” W), at 1840 m altitude. The agave was dissected in pine head (P), base of the scape (BS), base of the leaf (BL), and leaf (L) (Appendix A). The different parts of the agave were reduced in size to obtain pieces approximately 1.5 × 8 cm. After that, the juice extracted from each part was obtained in a commercial Rudo juice extractor TU05 (Turmix^®^, Mexico) with 440 W of power. The agave juices were labeled as P-juice, BS-juice, BL-juice, and L-juice. All juices were vacuum filtered through 0.22 µm membranes. After that, each juice sample was mixed for 15 min and then divided into aliquots. The aliquots were used for physicochemical analysis and fermentation assays. The juices for physicochemical analysis were poured into 1 L polypropylene containers using a maximum of ¾ of the volume and stored at −20 °C in an Ultra FreezerMDFU-5586SC-PA (Panasonic^®^, Mexico) until analysis. The juices for fermentation assays were poured into 1 L media bottles with a cap (KIMBLE^®^ KIMAX^®^, Germany) using a maximum of ¾ of the volume, and they were immediately sterilized at 121 °C in an Autoclave FE-399 (Felisa^®^, Mexico) for 15 min. After that, the bottles were completely closed, and when the bottles reached room temperature, they were stored at −4 °C until the experiments.

### 2.2. Microorganisms

The study used fifteen different yeasts available in our laboratory: (1) the USDA collection and (2) the strain collection of TecNM/Instituto Tecnológico de Cd. Hidalgo, and (3) the collection of strains from TecNM/Instituto Tecnológico de Morelia, all of which were isolated from diverse and different sources. The yeasts used in this study were *Candida cylindracea* NRRL Y-17537, *Lachancea thermotolerans* NRRL Y-2231, *Torulaspora delbrueckii* NRRL Y-1535, and *Yarrowia lipolytica* NRRL Y-5386 from the USDA collection. *Issatchenkia terricola* Y14 and *Pichia kluyveri* Y13 belong to the strain collection of TecNM/Instituto Tecnológico de Cd. Hidalgo. *Kluyveromyces marxianus* CDBB-L2029, *Pichia stipitis* ITMLB05, *Zygosaccharomyces bailii* ITMLB31, *Candida lusitaniae* ITMLB85, *Candida lusitaniae* ITMLB103, *Kluyveromyces marxianus* ITMLB106, *Pichia kudriavzevii* ITMLB97, *Saccharomyces cerevisiae* ITMLB69, and *Saccharomyces cerevisiae* ITMLB70 belong to the strain collection of TecNM/Instituto Tecnológico de Morelia. Specifically, within the strain collection of TecNM/Technological Institute of Morelia are *C. lusitaniae* ITMLB85, *K. marxianus* ITMLB106, and *P. kudriavzevii* ITMLB97, which were previously isolated from *Agave mapisaga *by the working group. The yeasts were characterized via microbiological and molecular techniques, including DNA extraction, PCR amplification with primers ITS1 and ITS4, and restriction fragment length polymorphism (RFLP) analysis, from which the purified products were sequenced and aligned using the Molecular Evolutionary Genetics Analysis (MEGA, V.11). They were subsequently analyzed via the Basic Local Alignment Search Tool (BLAST) and the National Center for Biotechnology Information (NCBI). The strains were maintained on YPD agar at 4 °C.

### 2.3. Physicochemical Analysis of the Agave Juices

The juices were subjected to the following determinations in triplicate: pH (pH meter, Hanna Instruments HI2211, Woonsocket, Rhode Island, USA), density (pycnometer, Kimble Glass Inc., USA), soluble solids (Handheld refractometer, LUZEREN SD090, Mexico), humidity (NMX-F-83-1986), ash (NOM-F-66-S), protein (Bradford), total phenolic content (Folin–Ciocalteu), and thin layer chromatography (TLC) [39] using chicory FOS as a commercial FOS reference sample (Megazyme). The data were analyzed by ANOVA and Tukey tests with *α =* 0.05 and *n* = 3.

### 2.4. Yeast Growth Screening in Different Agave Juices

The growth of the fifteen yeast strains (*C. cylindracea* NRRL Y-17537, *C. lusitaniae* ITMLB85, *C. lusitaniae* ITMLB103, *I. terricola* Y14, *K. marxianus* CDBB-L2029, *K. marxianus* ITMLB106, *L. thermotolerans* NRRL Y-2231, *P. kluyveri* Y13, *P. kudriavzevii* ITMLB97, *P. stipitis* ITMLB05, *S. cerevisiae* ITMLB69, *S. cerevisiae* ITMLB70, *T. delbrueckii* NRRL Y-1535, *Y. lipolytica* NRRL Y-5386, and *Z. bailii* ITMLB31) was evaluated in all the agave juices (P, BS, BL, and L). For this purpose, an independent preinoculation of each culture was carried out with two microbiological loops of the corresponding yeast, which were seeded in 250 mL Erlenmeyer flasks with 100 mL of YPD medium (20 g/L casein peptone, 20 g/L dextrose, and 10 g/L yeast extract) and incubated for 24 h at 30 °C and 150 rpm. The cellular concentration was subsequently determined in a Neubauer chamber. For that, 100 µL sample of the corresponding medium was taken and mixed with 10 µL of methylene blue and 890 µL of distilled water. From this mixture, 10 µL was added to each grid of the Neubauer chamber. On this basis, the volume of preinoculum required to obtain inoculums of 3 × 10^6^ cells/mL was determined. These inoculums were subsequently seeded in 50 mL Falcon tubes with 25 mL of the corresponding agave juice (P, BS, BL, or L) in triplicate. The growth of all yeasts was tested at different temperatures (25, 35, and 45 °C), and the pH was 5.5. The pH in all the fermentations was adjusted by adding 0.1 N hydrochloric acid (HCl) or 0.1 N sodium hydroxide (NaOH) if necessary. In addition, the growth of yeasts was monitored and quantified every 6 h for 30 h. These assays were used to select the juice that would serve as a substrate for the yeasts, as well as the yeasts for the following experiments.

### 2.5. Effects of Sucrose Concentration on FOS Production

To determine the effect of sucrose concentration on FOS production, BS-juice was selected as the medium for the growth of yeasts because it was the source where the yeasts grew most abundantly. For this purpose, different sucrose concentrations (15, 200, or 400 g/L) were evaluated with three yeasts selected because they grew in the BS-juice. In addition, these yeasts belong to different species and were all isolated from *A. mapisaga* and molecularly characterized in previous works of the research group: *K. marxianus* ITMLB106, *P. kudriavzevii* ITMLB97, and *C. lusitaniae* ITMLB85. Notably, the BS-juice used in all the fermentations was adjusted to 8° Brix with sterile distilled water. To determine the effect of sucrose concentration on FOS production, preinocula were prepared in 250 mL Erlenmeyer flasks with 100 mL of YPDE (20 g/L casein peptone, 20 g/L dextrose, 10 g/L yeast extract, 15 g/L sucrose, 1 g/L K_2_HPO_4_, 2.3 g/L KH_2_PO_4_, 1 g/L (NH_4_)NO_3_, 1 g/L (NH_4_)_2_HPO_4_, and 0.5 g/L MgSO_4_), to which two microbiological loops of the corresponding yeast were added. These flasks were incubated for 16 h at 30 °C and 150 rpm. The cellular concentration was subsequently determined in a Neubauer chamber, and inoculums of 3 × 10^6^ cells/mL of each yeast were inoculated in 250 mL Erlenmeyer flasks with 100 mL of enriched BS-juice (1 g/L K_2_HPO_4_, 2.3 g/L KH_2_PO_4_, 1 g/L (NH_4_)NO_3_, 1 g/L (NH_4_)_2_HPO_4_, and 0.5 g/L MgSO_4_), which, in addition, contained the respective sucrose concentration (15, 200, or 400 g/L). The inoculated flasks were subsequently incubated at 23, 30, or 37 °C for 56 h at 150 rpm, and the pH was 5.5 (adjusted with 0.1 N HCl or 0.1 N NaOH if necessary). Each treatment was performed in triplicate. Samples for cellular growth were taken at 0, 24, 48, and 56 h and immediately processed. The data were analyzed by ANOVA and Tukey tests with *α* = 0.05 and *n* = 3. In parallel, samples for TLC were taken and stored at −20 °C until analysis.

### 2.6. Effects of Surfactants on FOS Production

Previous studies reported that the use of surfactants at 10 mM promoted FOS production. Therefore, the effects of different surfactants (ionic and nonionic) were evaluated in the agave yeasts *P. kudriavzevii* ITMLB97 and *C. lusitaniae* ITMLB85. For this purpose, BS-juice was used in all fermentations and adjusted to 8° Brix with sterile distilled water.

The yeasts were grown in enriched BS-juice supplemented with 200 g/L sucrose and the corresponding surfactant at 10 mM: sodium deoxycholate (DNA), sodium dodecyl sulfate (SDS), Tween 80, or Triton X-100. The control treatments did not include surfactants. Previously, preinocula of *P. kudriavzevii* ITMLB97 and *C. lusitaniae* ITMLB85 were prepared in 250 mL Erlenmeyer flasks with 100 mL of YPDE, to which two microbiological loops of the corresponding yeast were added. These flasks were incubated for 16 h at 30 °C and 150 rpm. Then, the cellular concentration was determined in a Neubauer chamber, and inoculums of 3 × 10^6^ cells/mL were added to 50 mL Erlenmeyer flasks with 30 mL of enriched BS-juice supplemented with sucrose and the corresponding surfactant. After that, the inoculated flasks were incubated at 30 °C for 72 h at 150 rpm and pH 5.5 (adjusted with 0.1 N HCl or 0.1 N NaOH if necessary). Each treatment was performed in triplicate. Samples were taken at 0, 24, 48, and 72 h for analysis of cellular growth and were immediately processed. The data were analyzed by ANOVA and Tukey tests with *α* = 0.05 and *n* = 3. Additionally, samples for TLC were taken and stored at −20 °C until analysis.

### 2.7. Effects of Carbon Sources and Nutrients on FOS Production

Media with other carbon sources and nutrients were analyzed with *P. kudriavzevii* ITMLB97 or *C. lusitaniae* ITMLB85 to understand the type of enzymatic activity that produced the FOS previously observed. The treatments were performed in triplicate (Table 1). The different carbon sources used were chicory FOS (Megazyme), chicory inulin (SIGMA) and BS-juice; the commercial nutrients used from ININBIO were Di-phosta, Multicel, Forte, Nutri-fast, and Plus-cel. The general composition of the nutrient formulas consists of Di-phosta = assimilable nitrogen; Multicel = minerals such as potassium, sodium, magnesium, and calcium; organic nitrogen; Forte = minerals such as potassium, sodium, and ammonium salts as nitrogen sources; Nutri-fast = salts of ammonium, magnesium, calcium, and potassium; and Plus-cel = potassium, sodium, magnesium, and calcium. Previously, preinocula of *P. kudriavzevii* ITMLB97 and *C. lusitaniae* ITMLB85 were prepared in 250 mL Erlenmeyer flasks with 100 mL of YPDE, to which two microbiological loops of the corresponding yeast were added. These flasks were incubated for 16 h at 30 °C and 150 rpm. The cellular concentration was subsequently determined in a Neubauer chamber, and inoculums of 3 × 10^6^ cells/mL were added to 50 mL Erlenmeyer flasks containing 30 mL of DCSN medium (10 g/L yeast extract, 20 g/L casein peptone, 200 g/L sucrose, 13.33 g/L carbon source, and 0.25 g/L nutrient) according to Table 1. The inoculated flasks were incubated at 30 °C for 192 h at 150 rpm and pH = 5.5 (adjusted with 0.1 N HCl or 0.1 N NaOH if necessary). Each treatment was performed in triplicate. Samples for cellular growth were taken at 0, 24, 48, 120, and 192 h and were immediately analyzed. The growth at 192 h was analyzed by ANOVA and Tukey tests, with *α =* 0.05 and *n* = 3. In addition, samples for FT-IR (Fourier transform infrared spectroscopy), TLC (derivatized with α-naphthol) , and HPAEC-PAD (high-performance anion exchange chromatography) were taken and stored at −20 °C until analysis.

### 2.8. Fructosyltransferase Activity Evaluation

Box–Behnken designs were established to evaluate the fructosyltransferase activity of the enzymatic extracts of the yeasts *P. kudriavzevii* ITMLB97 and *C. lusitaniae* ITMLB85 obtained through two induction media. The first induction medium was reported by Chen *et al.* [23] for the induction of fructosyltransferase activity from an Ffase (3 g/L yeast extract, 5 g/L peptone, 30 g/L sucrose, and 0.5 g/L MgSO_4_·7H_2_O). The second proposed induction medium was NM (10 g/L yeast extract, 20 g/L peptone, 200 g/L sucrose, and 0.25 g/L diphosta nutrient from ININBIO). For this purpose, preinocula of each yeast were prepared in 250 mL Erlenmeyer flasks with 100 mL of YPDE, to which two microbiological loops of the corresponding yeast were added. These flasks were incubated for 16 h at 30 °C and 150 rpm. The cellular concentration was determined in a Neubauer chamber, and inoculums of 3 × 10^6^ cells/mL were subsequently added to 250 mL Erlenmeyer flasks with 100 mL of the corresponding induction medium, Chen (Ch) or NM. All the flasks containing the induction media were incubated at 30 °C for 48 h at 150 rpm. After incubation, the induction media were centrifuged at 8000 × *g* for 10 min, and the supernatants (crude extracts or enzymatic extracts) obtained from each induction medium were used to evaluate the enzymatic fructosyltransferase activity. These enzymatic extracts were named the following: the enzymatic extract of *P. kudriavzevii* ITMLB97 obtained from the induction medium of Chen (*EE-Pk-Ch*), the enzymatic extract of *P. kudriavzevii* ITMLB97 obtained from the induction medium of NM (*EE-Pk-NM*), the enzymatic extract of *C. lusitaniae* ITMLB85 obtained from the induction medium of Chen (*EE-Cl-Ch*), and the enzymatic extract of *C. lusitaniae* ITMLB85 obtained from the induction medium of NM (*EE-Cl-NM*) (Table 2). With each enzymatic extract, a Box–Behnken design was established in Statgraphics Centurion 18, in which three factors were evaluated: sucrose concentration (S = 200, 400, and 600 g/L), glucose concentration (G = 0, 300, and 600 g/L), and reaction time (t = 0, 2, 6, 12, and 24 h). All the enzymatic extracts were evaluated in 2 mL Eppendorf tubes with 1.5 mL of the reaction mixture composed of 1125 µL of sodium acetate buffer (pH = 5.5, adjusted with 0.1 N HCl or 0.1 NaOH if necessary) with the corresponding mixture of substrates (Table 2) and 375 µL of the corresponding enzymatic extract. The Eppendorf tubes with the reactions were incubated for 24 h at 50 °C and 550 rpm in a Thermomixer. In addition, 50 µL aliquots were taken at 0, 2, 6, 12, and 24 h for TLC (derivatized with α-naphthol) and HPAEC-PAD analyses. For the Box–Behnken designs, the response variables were the FOS detection signals (nC of 1-kestose (K), blastose (B), inulobiose (Ib), 6-kestose (6K), levanobiose (Lb), and neokestose (nK)).

### 2.9. Fourier Transform Infrared (FT-IR) Spectroscopy

A Thermo Scientific Nicolet™ iS50 FTIR spectrometer (Thermo Fisher Scientific, Madison, Wisconsin, USA) was used for FT-IR analysis. The instrument featured an attenuated total reflectance (ATR) diamond array. For analysis, 3 μL of each sample were taken during fermentation at 0, 24, 48, 120, and 192 h for each treatment in triplicate, and the samples were separately placed on the plate. FT-IR spectra of the samples were collected in the 4000–600 cm^−1^ region, and 32 scans were recorded at a nominal resolution of 4 cm^−1^ in transmission mode (%T). All the generated data were analyzed via principal component analysis (PCA) and orthogonal projections to latent structures (OPLS) via SIMCA-P (V.18) software. All the models were scaled by the unit variance (UV)-method. OPLS models were cross-validated via permutation tests (100 permutations) with *Q**^2^* ≥ 0.40 and CV-ANOVA tests with *p* < 0.05. To check the effects of different nutrient formulas in the IR profiles (composition) combined with inulin, FOS, and BS-juice as carbon sources, the nutrient formulas, used as mineral sources, were set as classes plus a control without any formula in a soft independent modeling of class analogy (SIMCA) analysis. To determine how similar or different the profiles were among each other, the control samples were set as the origin model. In this analysis, local PCAs are individually generated for each class, and the distances from each sample to each model are calculated and expressed as distance to the model (DmodX)-values. To distinguish the origin samples from other treated samples, a Dcrit-value of 0.05 was established; that is, everything below the Dcrit-value cannot be differentiated from the control samples. Conversely, all samples above the Dcrit-value can be differentiated from the control samples. Moreover, the differences between the treatment samples and the control samples were considered greater as their DmodX values increased. All the models were scaled via the unit variance (UV) scaling method [40]. The analyses were carried out in SIMCA-P (V.18, Umeå, Sweden) [40,41,42,43].

### 2.10. Thin Layer Chromatography (TLC) Analysis for Fermentations

For sample preparation, 200 μL of each sample was concentrated in an Eppendorf Vacufuge Plus Concentrator for 5 h at 30 °C. The samples were subsequently dissolved in 300 μL of water and sonicated. Then, 200 μL of absolute ethanol was added, and the samples were sonicated again. All the samples were prepared at 7 mg/mL, and 7 μL of each sample was applied to an TLC Silica gel 60 W aluminum plate (Merck, Darmstadt, Germany) with a CAMAG Automatic TLC sampler ATS4 (Muttenz, Switzerland) under the following conditions: injection speed: 8 μL/s; predosage: 200 nL; retraction volume: 20 nL; dosing speed: 70 nL/s; rinse/vacuum cycles: 1/8 s; fill/empty cycles: 1/1 s; application length: 6 mm; separation between bands: 10 mm; and seventeen bands for the sheet. For the standards, a total of 2 μL of each one was applied at 2 mg/mL (glucose, fructose, sucrose, 1-kestose, 1-nystose, and 1-F fructofuranosylnystose). In addition, 1-kestose, 1-nystose, and 1-F fructofuranosylnystose were used to identify some isomers with the same DP when TLC analyses were performed. The plates were developed in a biascendent separation. First, the chamber was saturated for 20 min in the isopropanol–butanol–water–acetic acid phase (14:10:4:2 *v*/*v*/*v*/*v*) and developed for 75 mm. After the first chromatographic development, the chamber was saturated for 20 min with a mixture of isopropanol–butanol–water–acetic acid–formic acid (14:10:4:1:1 *v*/*v*/*v*/*v*/*v*) and the plate was developed for 85 mm. The TLC plate was subsequently dried and derivatized with α-naphthol at 100 °C for 3 min. After that, images of the plates were recorded in a CAMAG TLC visualizer 2 [44].

### 2.11. High-Performance Anion-Exchange Chromatography with Pulsed Amperometric Detection (HPAEC-PAD) Analyses

The FOS profiles were obtained via a DIONEX ICS-3000 chromatograph (Thermo Fisher Scientific, Waltham, MA, USA) equipped with a precolumn (40 × 25 mm) and a Dionex CarboPac PA-200 column (40 × 250 mm, from Thermo Fisher Scientific, Sunnyvale, CA, USA) maintained at 25 °C. The elution solutions for sample separation were NaOH [0.23 M], CH_3_COONa [0.5 M], and water at a constant flow rate of 2 mL/min. All the samples were adjusted to 1.75 mg/mL, and all the samples, including the standards, were filtered through a 0.22 μm nylon membrane (Merck Millipore, Massachusetts, USA) before injection. Only Milli-Q water was injected between each sample to clean any residue in the injector, precolumn, and column. Standards of glucose, fructose, sucrose, 1-kestose, 1-nystose, and 1-F fructofuranosylnystose were injected for compound identification. In addition, some donated mixtures of disaccharides (blastose, inulobiose, and levanobiose) and FOS (6-kestose and neo-kestose) were also injected as reference compounds. All the results were expressed as nC vs. time. The time of analysis for each sample was 75 min. The detection conditions were those reported by Salomé-Abarca *et al*. [44].

## 3. Results and Discussion

### 3.1. Physicochemical Analysis

The analysis of the data obtained from physicochemical analyses revealed significant differences between the juices obtained from the different parts of the agave (Table 3). These findings suggest that, in general, there are changes in the composition of each part of the agave, depending on its function in the plant. The pH values ranged from 5.02 to 5.51, their density ranged from 1.183 to 1.0483 g/mL, and the °Brix ranged from 4.8 to 10.8 (Table 3). Interestingly, all the values increased from external (L and BL) to internal (BS and P) regions of the plant. In line with our results, aguamiel obtained from agave pine heads tends to possess acid pH [45]. For example, the pH values of *A. salmiana* and *Agave* spp. are 4.37 and 6.43, respectively [46,47]. However, aguamiel can also display soft alkali characteristics, such as those of *A. americana,* with a pH of 7.72 [48]. The density and °Brix values for the aguamiel of *A. americana* ranged between 1.034 and 1.055 g/mL and 10 °Brix, respectively [49]. These values were also in line with those observed in this study (10.8 °Brix). However, *°*Brix variation has been reported to occur between 9.3 and 16 °Brix in the aguamiel of *A. salmiana* and *Agave* spp.; this variation could be related to agave age differences [50,51]. In addition, physicochemical variations in agave sap depend on the agave species [29,30,52,53] and their growing environmental conditions [54].

The moisture percentage in the agave juices varied between 92.08% and 96.76%. The moisture content reported for the aguamiel of *A. atrovirens* was 89.61%, and that reported for the agave leaves was between 79.54 and 88.00% [45,55], indicating that the plant materials used in this study contained more water. The ash content in the juices used in this study ranged between 1.120 and 1.673% (Table 3). Other studies reported ash content in the aguamiel of *A. mapisaga* around 3.3 ± 0.08% [35]. Ash values indicate mineral content, including K, Ca, Na, Fe, Cu, Mg, Se, and Zn, which means that our agave materials possess approximately half the mineral content of *A. mapisaga* [56].

The protein content of the P-juice was greater (41.28 μg/mL) than that of the BS-, BL-, and L-juices. The protein content of P-juice was twice as high as that of L-juice. The protein percentage in the aguamiel of *A. atrovirens* is 3.5% (3.5 g/100 g) [46]. The use of the Lowry method resulted in values of 3.03 to 3.35% [57], which were superior to those reported in this study. The total phenolic content ranged from 2.06 to 3.80 μg/mL. The P- and BS-juices possessed the highest phenolic contents among all the agave juices (Table 3). Studies of the aguamiel of *A. atrovirens* reported 3.02 GAE/g, whereas 2.26 GAE/g was reported for the aguamiel of *A. salmiana* [46,58].

Finally, TLC analyses of all the agave juices revealed differences in their carbohydrate compositions (Figure 1). The P-juice contained simple sugars such as fructose and glucose, the disaccharide sucrose, FOS molecules from DP3 to DP7, and HDP-fructans localized at the sample application point. Interestingly, BS-juice was mainly composed of fructose, glucose and DP3 (1-kestose), whereas BL-juice and L-juice were mainly composed of glucose. These results are very interesting because it has been reported that the fructan DP of agaves increases as their age increases too. Studies of *Agave angustifolia* Haw. from San Esteban Amatlán, Oaxaca, Mexico, and *Agave potatorum* Zucc. from *Infiernillo, Zaachila, Oaxaca, Mexico* showed fructose and glucose as the most abundant carbohydrates in 1-year-old plants. Later, when the plant grows, between 1- and 3-year-old FOS are the most common carbohydrates, and after 4 years, HDP-fructans become predominant [59]. A similar behavior is reported for *Agave atrovirens* Karw. from Singuilucan, Hidalgo, Mexico [60]. Additionally, similar results were reported for *A.* spp., *A. atrovirens* and *A. mapisaga* spp. Crassipina from Perote, Veracruz, *A. tequilana* cenizo variety from Jalisco and *A. salmiana* chino variety from San Felipe, Guanajuato richer in FOS during their first four years and richer in HDP-fructans, up to DP-70 in 10- to 12-year-old specimens [61]. An important difference between these agaves is that they belong to different species and come from different regions. In this sense, it is interesting to ask ourselves the following questions: What could be the determining factor in observing such features? Could it be associated with environmental factors?

### 3.2. Yeast Growth Screening in Different Agave Juices

The ability of all yeasts to grow in all the agave juices (P, BS, BL, or L) was tested by using them as part of their culture media. At 30 h, all the tested yeasts did not grow in the BL- or L-juices (Figure 2). Even if TLC analyses revealed the presence of simple sugars in the BL- and L-juices, which might serve as substrates for yeast growth, there could be other metabolites in these tissues with antimicrobial properties. In plants, polyphenols, saponins, and terpenes have antimicrobial properties [62,63,64]. All these compounds have been reported to be common components of agave leaves, which have also been reported to display antimicrobial properties [65]. Studies carried out on the leaves of *Agave sisalana* have shown that its methanol and aqueous extracts inhibit the growth of microbes [66]. In addition, leaf ethanolic extracts of *A. lecheguilla* Torr., *A. picta* Salm-Dyck, *A. scabra* Salm-Dyck, and *A. lophantha* Schiede showed antimicrobial activity against *Cryptococcus neoformans* and other filamentous fungi [67]. Antibacterial potential has been reported for crude extracts of *A. americana* leaves, which was supported with phytochemical analysis, which revealed the presence of alkaloids, saponins, phenols, flavonoids, and tannins [68]. In the future, antimicrobial agents from Agave could be obtained from the least used organ, the agave leaves. On the other hand, *C. lusitaniae* ITMLB85, *K. marxianus* ITMLB106, and *P. kudriavzevii* ITMLB97 grew at 25 and 35 °C in P- and BS-juices. Nonetheless, these yeasts grew better in BS-juice, above 1 × 10^7^ cells/mL. Therefore, BS-juice, *C. lusitaniae* ITMLB85, *K. marxianus* ITMLB106, and *P. kudriavzevii* ITMLB97 were selected for subsequent tests.

**Figure 2 foods-14-02714-f002:**
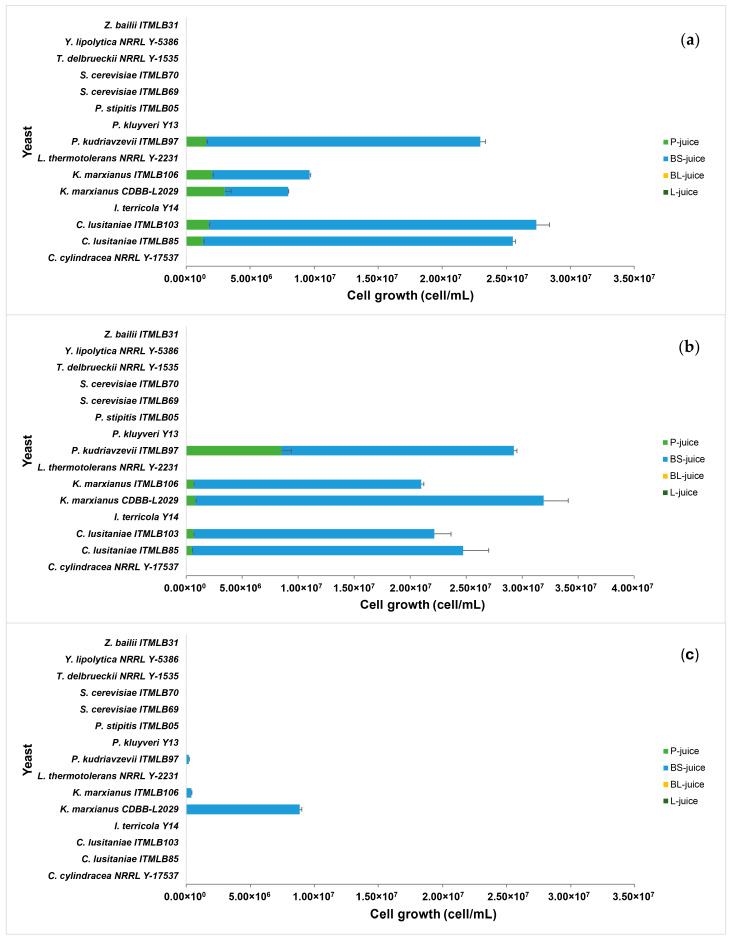
Yeast growth in agave juices (**P** = pine head juice, **BS** = base of scape juice, **BL** = base of leaf juice, and **L** = leaf juice) at 30 h at different temperatures: (**a**) 25 °C, (**b**) 35 °C, and (**c**) 45 °C. The color in the graphics represents the cell growth in the corresponding juice; in some cases, the color is not visible due to the absence of growth. Each bar represents the average of three replicates.

### 3.3. Effects of Sucrose on FOS Production

FOS production is associated with sucrose concentration, growth temperature, and the yeast species. From individual growths of *K. marxianus* ITMLB106, *C. lusitaniae* ITMLB85, and *P. kudriavzevii* ITMLB97, it was determined that the three yeasts were able to grow with all the tested sucrose concentrations in the media (Figure 3 and Appendix A). However, a growth delay was observed as the sucrose concentration increased from 1.5 to 40% (Figure 3a–c). Through ANOVA and Tukey’s tests (α = 0.05) at 56 h, significant differences were found between the treatments of each yeast (Figure 3a–c).

In the case of *C. lusitaniae* ITMLB85 and *P. kudriavzevii* ITMLB97, the sucrose concentration and temperature had important effects on FOS production, especially for 20% sucrose at 30 and 37 °C, where a specific increase in 1-nystose (DP4) was observed (Figure 3e,f,h,i). Gomes-Barbosa *et al*. [69] reported that increasing the sucrose concentration in the medium, above 10%, resulted in the increase in fructosyltransferase activity of the invertase from *R. mucilaginosa* and *S. cerevisiae*. Moreover, Muñiz-Marquez *et al.* [70] reported that Ffase enzymes could display transfructosylation activity but only under relatively high sucrose concentrations in the medium. There was small growth in the case of *K. marxianus* ITMLB106 (Figure 3b) and showed only glucose, fructose, and sucrose degradation but not FOS production. Thus, only *C. lusitaniae* ITMLB85 and *P. kudriavzevii* ITMLB97 were selected for further analyses. However, as the increase in FOS production in these two yeasts was not that high, other FOS production stimulant agents were further investigated.

### 3.4. Surfactant Effects on FOS Production

The use of surfactants was reported as a strategy to increase the production of FOS [71]. Thus, different surfactants, including SDS, DNa, Tween 80, and Triton X-100, were evaluated to increase the fructosyltransferase activity of *P. kudriavzevii* ITMLB97 and *C. lusitaniae* ITMLB85. The first two are considered ionic surfactants, while Tween 80 and Triton X-100 are nonionic. The cell growth of *P. kudriavzevii* ITMLB97 and *C. lusitaniae* ITMLB85 in response to surfactants is shown in Figure 4 and Appendix A. Both yeasts did not grow and died when the media with BS-juice contained SDS [10 mM], suggesting that this SDS concentration induced cellular lysis due to pore opening on the lipidic membrane of the yeast (Figure 4a,b). This was also observed on the TLC profile of the yeast growing in this treatment, where no profile change was observed due to the death of the yeasts (Figure 4c,d). This was also confirmed under the microscope. Moreover, all the treatments, except SDS, caused the consumption of glucose and fructose in the first 24 h by both microorganisms (Figure 4c–e).

Conversely, the treatment with DNa in *P. kudriavzevii* ITMLB97 (Figure 4c) and the treatment with Triton X-100 in *C. lusitaniae* ITMLB85 (Figure 4d) showed a clear increase in their FOS production over time. This increase occurred in DP3, DP4, DP5, and DP6, but it was more notable in DP4. Interestingly, the TLC profile of pure BS-juice showed mostly DP3, which supports the production of FOS under these conditions. The use of surfactants in culture media affects the physiology of yeasts [72] by modifying the permeability of membranes, which allows free mobility of substrates and products across cell membranes [73]. The capability of cell permeabilization to form pores depends on the nature of the surfactant. Studies performed with a nonionic surfactant (Triton X-100) revealed an initial increase in the surface area of the bilayer due to the insertion of the surfactant between the two layers, which generated several open holes. Differently, the ionic surfactant SDS revealed an increase in the membrane curvature associated with the incorporation of the surfactant only in the outer layer, which resulted in the opening of macropores [74,75]. Additionally, the permeation effects depend on the membrane sterol content [72,76], which varies depending on the yeast species, which might explain why different surfactants are better for different yeast species. With these results in hand, the next step was to test different carbon sources and nutrients to improve the production of FOS.

### 3.5. Effects of Carbon Sources and Nutrients on Yeast Growth and FOS Production

The fermentation of different treatments with *P. kudriavzevii* ITMLB97 or *C. lusitaniae* ITMLB85, which use distinct carbon sources (FOS, inulin, and BS-juice) and diverse nutrients (Di-phosta, Multicel, Forte, Nutri-fast and Plus-cel), showed in Table 1, resulted in different cell growth rates (Figure 5 and Appendix A). In the case of *P. kudriavzevii* ITMLB97, the use of FOS as a carbon source combined with the Di-phosta nutrient (*Pk-FOS-di*) increased the yeast growth in the first 48 h. The nutrient Di-phosta is composed of diammonium diphosphate ((NH_4_)_2_HPO_4_), which, in previous studies with *S. cerevisiae*, has been reported to be an efficient nutrient for increasing the yeast population [77,78] (Figure 5a). On the other hand, when *P. kudriavzevii* ITMLB97 used inulin as a carbon source with Multicel nutrient (*Pk-inulin-multi*), the cellular growth increased during the first 48 h (Figure 5c). This could be related to the Multicel nutrient composition, including potassium, sodium, magnesium and calcium, which are essential for cells. Intracellular potassium and sodium are vital for cell growth and cellular functions to maintain electroneutrality, proper membrane potential and intracellular pH, as well as cell turgor and volume, protein synthesis, and enzyme activity. In addition, magnesium is involved in several physiological functions, such as growth, cell division, enzyme activity, and structural stabilization of nucleic acids, polynucleotides, chromosomes, polysaccharides, and lipids, as well as ribosome structure [79,80,81]. The addition of Mg^2+^ to media stabilizes biological membranes and is known for protecting yeast cells from stress caused by ethanol, temperature, and osmotic pressure [82]. On the other hand, Ca^2+^ antagonizes several Mg^2+-^dependent functions of yeast, such as growth and metabolism, through inhibitory competitive binding mechanisms [82]. Calcium is also needed by yeasts at concentrations ranging from 0.25 to 0.5 mM [83]. In the case of *C. lusitaniae* ITMLB85, when BS-juice was used as carbon source, the most noticeable effect on cellular growth was observed for the Di-phosta nutrient (*Cl-BS-di*) (Figure 5f). Park *et al*. [84] reported that ammonium ions could reduce the lag period of the growth kinetic during FOS production.

Even if there are specific treatments with enhanced yeast growth, fructan production must be profiled. In this context, other analytical platforms that can provide deeper structural information should be employed. Thus, the first screening of the culture media of all the treatments was scrutinized by FT-IR in the MIR range (600–4000 cm^−1^). The spectra were processed and subjected to multivariate data analysis via PCA. The datasets from *P. kudriavzevii* ITMLB97 and *C. lusitaniae* ITMLB85 were independently analyzed. In the case of *P. kudriavzevii* ITMLB97, the model produced 15 principal components (PCs), which explained 99% of the total variation in the model (RX^2^cum = 0.99). The samples were separated along PC1 and PC2, which captured 41 and 32% of the variation in the model, respectively. This analysis revealed that the main factor influencing sample separation was the carbon source in the culture media (Figure 6a). These findings indicate that the carbohydrates contained in each carbon source are differentially metabolized by *P. kudriavzevii* ITMLB97. This difference might be associated with differences in fructan type and DP. For example, inulin and FOS are linear β(2→1)-fructans, whereas BS-juice contains β(2→1/2→6)-branched fructans. Furthermore, FOS were restricted to low-DP fructans. In this context, fructans with low DP are rapidly fermented, which results in a better prebiotic effect [42]. This can be corroborated when examining the growth values of controls of *P. kudriavzevii* ITMLB97, where FOS produced the highest growth values in this yeast (Figure 5). Similar results were also observed when the *C. lusitaniae* ITMLB85 data were analyzed via PCA. In this case, the model needed only five PCs to explain 95% of the total variation in the dataset (RX^2^cum = 0.95). The model also needed three PCs to properly separate the samples by carbon source; however, this model better separated the FOS, inulin, and BS-juice clusters, mainly in PC3 (Figure 6b). This clearer separation between the FOS- and inulin-treated samples might indicate that *C. lusitaniae* ITMLB85 metabolizes fructans in a different way than *P. kudriavzevii* ITMLB97. For example, the better separation between inulin and FOS indicates that *C. lusitaniae* ITMLB85 is more sensitive to changes in the DP of the carbon source. That is, changing fructan DPs results in different metabolic products being excreted into the culture medium, which ultimately results in a clearer separation between these two treatments. In addition, *C. lusitaniae* ITMLB85 was more sensitive than *P. kudriavzevii* ITMLB97 to fructan type since branched agavins also caused better separation of its cluster. In this context, even if the type and DP of fructans play crucial roles in the development of microorganisms, it has been reported that the microorganism species is also a determining factor dictating the microorganism–fructan interaction outcome. For example, different species of *Bifidobacterium* and *Lactobacillus* showed four different growth patterns that depended more on the bacterial species rather than on the DP of fructans [43].

In addition to carbon sources, other factors could influence the separation of all the treatments, that is, influencing the differentiation of their metabolism in response to different types of fructan. For instance, the combination of carbon sources with different nutrient formulas. When a PCA was inspected to look for such effects, there was no clear subclustering according to nutrients. However, this might be caused by parallel factor variation affecting sample classification. In the context, when dealing with several factors affecting the same dataset, unsupervised analyses are not enough to characterize the individual effects of each factor. Thus, supervised methods can suppress undesirable effects, which clarify the effects of the target factor [41], in this case, nutrient effects. Therefore, to check the effects of different nutrient formulas in the IR profiles (composition) combined with inulin, FOS, and BS-juice as carbon sources, the nutrient formulas were set as classes plus a control without any nutrient formula in a soft independent modeling of class analogy (SIMCA) analysis. For *P. kudriavzevii* ITMLB97, the analyses revealed that the different nutrients caused changes in the infrared profile of the treatments since they differed from those of the control (Appendix A). The highest DmodX values for inulin and FOS indicate that the combination of inulin and FOS with the different nutrients caused greater changes in the composition of the media than the combination of BS-juice and the different nutrients. However, compared with the other combinations, the Plus nutrient formula consistently caused changes of lesser magnitude in the medium composition, which was supported by its lower DmodX values. Additionally, the Di-phosta, Multicel, and Forte formulas always caused the most significant changes when combined with any of the three carbon sources, although these effects were more subtle when the carbon source was BS-juice, where the DmodX values of the Nutri formula were more similar to those of the other treatments. This could be related to the fact of BS-juice represents a medium that is more similar to its natural development environment in planta.

For *C. lusitaniae* ITMLB85, the analysis revealed that the different nutrients caused changes in the infrared profile of the treatments since they differed from their respective controls (Appendix A). However, the different formulas showed highly contrasting behaviors depending on what carbon source they were combined with. In this yeast, the Plus formula was the least differentiated, as in the case of *P. kudriavzevii* ITMLB97, but only when it was combined with FOS or BS-juice. In contrast, when combined with inulin, the Plus formula was clearly differentiated from the control, and its differentiation (DmodX values) increased along with its incubation time. This was observed as a gradual increase in the DmodX value from left to right in the samples subjected to this treatment. This could indicate that, over time, *C. lusitaniae* ITMLB85 metabolizes the carbon source and nutrients, which alters the composition of the culture medium, making it more different from the control culture medium. In this case, it was generally observed that the Di-phosta formula caused the greatest degree of differentiation from the control. Additionally, BS-juice combined with the different formulas resulted in the lowest degree of differentiation compared with the combinations of inulin or FOS with the different nutrients.

To further characterize and determine which infrared signals were correlated with each yeast species and their carbon source and nutrients, orthogonal projections to latent structure models were built. For this purpose, the MIR data were set as “X” data, whereas the incubation time were set as quantitative “Y” data. All the analyses revealed a correlation between MIR variation and culturing time (Figure 6c–h). All the models were well validated (*Q^2^* > 0.40 and *p* < 0.05). In the case of *P. kudriavzevii* ITMLB97, the best correlation models were those produced from inulin (*Q^2^* = 0.93) and BS-juice (*Q^2^* = 0.93), which had a highly significant effect (*p* < 0.0001) on the MIR profiles. On the other hand, the FOS model was also validated, but it produced a lower *Q^2^* (0.79) and *p* value (0.01); however, it was still validated. Similarly, for *C. lusitaniae* ITMLB85, the FOS model produced a *Q^2^* of 0.55, whereas the inulin and BS-juice models produced *Q^2^* values of 0.95 and 0.94, respectively. These two models also yielded a *p* < 0.0001. These results indicated that the composition of the culture medium linearly changed through their culturing time. To gain more insight into what was correlated with such changes, a VIPpred plot was produced from each OPLS model. For *P. kudriavzevii* ITMLB97 fed FOS, the most correlated signals included wavenumbers between 975 and 985 cm^−1^ and signals at approximately 831 cm^−1^. The same MIR range was determined for the inulin model plus vibrations at 1335–1337 cm^−1^. For the BS-juice model, the bands at 975–985 cm^−1^ were also correlated signals plus signals at approximately 800 cm^−1^. The region conserved in the three models is close to bands associated with *α-* and *β*-anomers in cyclic carbohydrates [85], whereas signals at approximately 831 cm^−1^ have been attributed to the *α*-configuration [86] but also to free fructose [87]. The signals near 1340 cm^−1^ are correlated with C–H bending vibrations [88], probably in the skeletons of carbohydrates. In the case of *C. lusitaniae* ITMLB85, the wave numbers most correlated with FOS feeding over time were approximately 830 and 780 cm^−1^ related to free fructose and the *α*-configuration, which is in line with the FOS model of *P. kudriavzevii* ITMLB97. These findings suggest that FOS are metabolized in a similar way by both yeast species. Conversely, the inulin and BS-juice models presented wavenumbers between 913 and 923 cm^−1^, and the region between 870 and 877 cm^−1^ was distinctive for the BS-juice model. Bands at approximately 920 cm^−1^ have been detected in diverse polysaccharides, which strongly suggests changes in the fructan composition in the culture media [44]. These results indicate that even if *P. kudriavzevii* ITMLB97 and *C. lusitaniae* ITMLB85 similarly metabolize FOS, they metabolize inulin and agavins [89] differently. Thus, different carbon source metabolism products might result from the combination of different carbon sources and microorganisms. Such differences can be associated with structural (qualitative), quantitative or DP changes. Nonetheless, to confirm such changes, a more informative analytical platform is needed. Thus, the next step of this research consisted of the analysis of the samples via HPAEC-PAD.

The HPAEC-PAD chromatograms obtained from the fermentations of different treatments including *P. kudriavzevii* ITMLB97 or *C. lusitaniae* ITMLB85 using distinct carbon sources (FOS, inulin and BS-juice) and diverse nutrients, such as Di-phosta, Multicel, Forte, Nutri-fast and Plus-cel, showed variation in their specific carbohydrate profiles (Table 1, Figure 7). In the treatments where FOS were the carbon source for *P. kudriavzevii* ITMLB97, the most important product found was blastose in the *Pk-FOS-di, Pk-FOS-forte, Pk-FOS-nutri*, and *Pk-FOS-control* treatments. In addition, 6-kestose was produced in the control treatment. On the other hand, when FOS were the carbon source in all the treatments for *C. lusitaniae* ITMLB85, blastose was the most common product. The *Cl-FOS-control* treatment improved FOS production, specifically 1-kestose, blastose, 6-kestose, 1-nystose, 1-F fructofuranosylnystose (DP5), and FOS with DP > 5. *C. lusitaniae* ITMLB85 may have potential fructosyltransferase activity for blastose- and ^1^F-FOS production. In addition, it is important to highlight that blastose found in the ferments is a disaccharide, which could be associated with fructose transfructosylation [17]. To our knowledge, the first report of blastose production by a Ffase of yeast, which involves the formation of this difructan by direct fructosylation of glucose was reported by Piedrabuena *et al*. [90]. However, blastose production is usually associated with neo-kestose hydrolysis [91], but in this case, neo-kestose was not present in the media. On the other hand, 1-kestose and 6-kestose are trisaccharides, which differ in the type of bond in their structure and therefore in the type of enzyme activity that forms them. In contrast to plants, yeasts can produce different DP-FOS via the same enzyme [8]. However, to produce 1-kestose, specific 1-SST (sucrose:sucrose 1-fructosyltransferase) fructosyltransferase activity is needed, whereas for 6-kestose, 6-SST (sucrose:sucrose 6-fructosyltransferase) fructosyltransferase activity is mandatory. Finally, 1-nystose, 1-F fructofuranosylnystose (DP5) and the FOS with DPs > 5 could be associated with 1-FFT (fructan:fructan 1-fructosyltransferase) fructosyltransferase activity. For example, 1- and 6-kestose have been reported to be produced by *Rhodotorula dairenensis* and *S. cerevisiae* [18,19]. Tetrasaccharides are produced by *R. dairenensis* [18]. Interestingly, the production of 1-nystose without 1-kestose has been reported in *S. cerevisiae* CAT-1 and *Rhodotorula mucilaginosa* [69].

When inulin was used as carbon source for *P. kudriavzevii* ITMLB97, the most abundant product was 6-kestose, followed by 1-kestose and blastose in the *Pk-inulin-di*, *Pk-inulin-nutri*, and *Pk-inulin-control* treatments. On the other hand, when inulin was used as carbon source for *C. lusitaniae* ITMLB85, 6-kestose and 1-kestose were also detected in the *Cl-inulin-control* treatment. In addition, 1-kestose was also observed with *Cl-inulin-Forte* treatment. Interestingly, in all the assays where fructosyltransferase activity was observed at 192 h, high quantities of glucose and fructose were detected. This result led to the question of whether Ffases under the conditions of these study required glucose and fructose to produce FOS. In this context, Raga-Carbajal *et al.* (2018) described that the production of the disaccharides blastose, levanobiose, and inulobiose was associated with the rapid accumulation of glucose and fructose [17]. On the other hand, in the treatments where inulin was used as a carbon source, possible inulinase activity was discarded since no degradation of fructans with a high degree of polymerization was observed. Furthermore, the products generated by inulinases include inulotriose, inulotetrose, and inulopentose, not present in our samples, and inulinases do not possess invertase activity to hydrolyze sucrose [8].

Moreover, the results when BS-juice was used as a carbon source were surprising because few peaks corresponding to 6-kestose were found with the *Pk-BS-nutri, Cl-BS-multi* and *Cl-BS-plus* treatments. The Ffase of the yeast *S. occidentalis* has been reported to be a good producer of 6-kestose [12,20].

### 3.6. Fructosyltransferase Enzyme Activity

Disaccharides and FOS production were detected in the HPAEC-PAD chromatograms with the different treatments to evaluate the enzymatic extracts of *P. kudriavzevii* ITMLB97 and *C. lusitaniae* ITMLB85 at 12 h (Table 2). In the case of *P. kudriavzevii* ITMLB97, both enzymatic extracts (*EE-Pk-Ch* and *EE-Pk-NM*) produced blastose, 1-kestose, and 6-kestose. However, the presence of 6-kestose notably increased with *EE-Pk-NM* at 12 h, specifically with the 3^Pk-NM^ and 9^Pk-NM^ treatments (Figure 8a,b). On the other hand, the enzymatic extracts of *C. lusitaniae* ITMLB85 (*EE-Cl-Ch* and *EE-Cl-NM*) also produced blastose, 1-kestose and 6-kestose. With *EE-Cl-Ch* 6-kestose was detected mainly in the 5^Cl-Ch^ treatment. However, more 6-kestose and an unknown fructooligosaccharide between DP5-DP6, were observed with *EE-Cl-NM* in the 3^Cl-NM^, 7^Cl-NM^, and 9^Cl-NM^ treatments. The 7^Cl-NM^ treatment also produced neo-kestose (Figure 8c,d). The production of neo-kestose is associated with ^6^G-FFT (fructan:fructan ^6^G-fructosyltransferase) fructosyltransferase activity. All these treatments had a minimum sucrose concentration of 200 g/L, corroborating the importance of sucrose concentration for fructosyltransferase activity. It is important to highlight that most of the media showed relatively high glucose content, which also suggests that its concentration plays a role in the function of yeast enzymatic extracts with fructosyltransferase activity in the formation of disaccharides and FOS.

Furthermore, the comparison between the profiles of the products formed by the enzymatic extracts of *P. kudriavzevii* ITMLB97 (*EE-Cl-Ch* and *EE-Cl-NM*) and *C. lusitaniae* ITMLB85 (*EE-Cl-Ch* and *EE-Cl-NM*) showed that the production of such products increased when the extracts were obtained with the NM induction medium. The differences between the induction media (Ch and NM) were determinants for disaccharides and FOS production. Differences between the Ch and NM induction media realies on their yeast extract, peptone, and sucrose contents, as well as the use of distinct salts. For instance, the Ch induction medium contained MgSO_4_·7H_2_O, whereas the NM induction medium contained Di-phosta nutrient (NH_4_)_2_HPO_4_). These findings suggest that ammonium could increase the fructosyltransferase activity. This phenomenon has previously been observed as a 15-fold increase in fructosyltransferase activity [84]. In addition, phosphate is one of the most important nutrients for microorganisms, not only because it is the main source of free energy required for many cellular processes but also because it forms parts of nucleotides, phospholipids, and nucleic acids. All this could promote the fructosyltransferase activity of the enzymatic extracts from the NM induction media.

Similarly, another important difference is that the fructosyltransferase activity of the enzymatic extracts varies depending on the yeast used, resulting in greater FOS production with *C. lusitaniae* ITMLB85 (Figure 8b,d). There are studies of fructosyltransferase activity in yeasts, among which the fructosyltransferase activity of *Candida* has been reported [92,93]. Therefore, the use of yeasts for disaccharides and FOS production could be an area to explore to obtain prebiotics.

The factors to evaluate in the treatments established by the Box–Behnken designs for the different induction media of *P. kudriavzevii* ITMLB97 or *C. lusitaniae* ITMLB85 were sucrose concentration (S), glucose concentration (G), and time. All the data generated were analyzed via HPAEC-PAD (Table 2), and all the signals generated by the ICS-3000 in the FOS region were analyzed. The RSs obtained are shown in Figure 9, where each column represents an enzymatic extract and each row represents a disaccharide or an fructooligosaccharide. The optimal conditions according to the scope and limitations of this study were as follows: 1-kestose (S = 385.84 g/L and G = 600 g/L at 24 h) with *EE-Cl-Ch* and inulobiose (S = 600 g/L and G = 600 g/L at 21.90 h) with *EE-Pk-NM*. The prebiotic effect of 1-kestose is well known. For example, its administration to obese people had important effects on the serum insulin level and increased the relative abundance of fecal *Bifidobacterium*. Therefore, 1-kestose improves insulin resistance in overweight humans via the modulation of the gut microbiota [94]. In addition, 1-kestose administration to elderly patients with sarcopenia increased the *Bifidobacterium longum* population, and after three months, the skeletal muscle mass index increased, and the body fat percentage decreased [95]. On the other hand, the induction medium with the greatest impact on disaccharides and FOS production was *EE-Cl-NM,* with the optimal production of the disaccharides blastose (S = 200.20 g/L and G = 600 g/L at 24) and levanobiose (S = 596.08 g/L and G = 599.90 g/L at 24 h) and the FOS 6-kestose (S = 200 g/L and G = 600 g/L at 10.35 h) and neo-kestose (S = 600 g/L and G = 0 g/L at 11.23 h). In addition, 6-kestose was the main product formed by the fructosyltransferase activity (6-SST) of *C. lusitaniae* ITMLB85, followed by blastose. Therefore, its fructosyltransferase activity can form β(2→6) bonds. The difference between these products is that when fructosylation occurs to produce 6-kestose, the fructosyl is linked to the 6-carbon of the fructose in the sucrose molecule, and when blastose is formed, the fructosyl is linked to the 6-carbon of the glucose. Both molecules have a β(2→6) link in their structure, which could be associated with improved effects on biological functions and increased prebiotic potential [12,18]. In addition, blastose is considered the base of blasto-FOS and neo-FOS [15,17]. Studies with blastose revealed that its coadministration with sucrose suppressed the increase in plasma glucose (PG). While neo-kestose suppresses melanoma cell viability via inhibition of the NF-κB pathway, neo-kestose also induces apoptotic cell death in colorectal carcinoma cells [96,97]. In addition, 6-kestose is the first synthetized molecule of the ^6^F-FOS, which promotes the growth of *Bifidobacterium lactis* and probably induces an increase in synthesis of short-chain fatty acids (SCFAs), reflected in the decrease in pH in the culturing medium [98]. However, it is necessary to focus studies on 6-kestose biological functions [99]. The efficient production of prebiotic molecules is necessary as a tool to modulate the intestinal microbiome and combat some critical diseases that affect the whole world.

## 4. Conclusions

Juices obtained from *Agave* sp. grown at Cuitzeo Lake in Michoacán display unusual DP compared with other reported agave juices. This might be related to the distinctive geographical features of the region; however, other studies must be performed. In addition, the physicochemical variation of agave juices depends on their origin, which impacts their suitability to serve as a growing medium for yeasts. BS-juice can be used as a yeast culture medium to produce blastose and 6-kestose, both of which bear potential health benefits. However, nowadays, it is impossible to consider its industrial production and other conditions as several agaves and yeasts species must be tested. On the other hand, L- and BL-juices are not recommended as yeast culture media. However, studies on their antimicrobial potential could be performed. In particular, *Agave* sp. represents a vast compendium of diverse species underused in the region of Michoacán, Mexico; therefore, they represent raw material with the potential to obtain bioproducts of interest, such as prebiotics, including ^1^F-FOS, ^6^F-FOS, and blasto-FOS.

The sucrose concentration (20%) and temperature are determining factors in observing yeast fructosyltransferase activity; however, increasing the sucrose content in the medium to 40%, as well as the temperature (37 °C), did not result in improved FOS production under the conditions studied.

In general, the surfactants DNa, Tween 80, and Triton X-100 at 10 mM maintain or promote yeasts’ fructosyltransferase activity for FOS production. However, depending on the yeast and the surfactant, the surfactant concentration must be evaluated because yeast cells could be lysed as occurred with 10 mM SDS in this study. From the assays with different carbon sources and nutrients with *P. kudriavzevii* ITMLB97 or *C. lusitaniae* ITMLB85, it was determined that although the use of commercial nutrients promotes cell growth, it is not associated with increases in FOS production. Cell growth was promoted in some cases, for example, for *P. kudriavzevii* ITMLB97 with Di-phosta and Multicel nutrients, and for *C. lusitaniae* ITMLB85 with the Di-phosta nutrient. In addition, from the assays with different carbon sources and nutrients, where all the experiments also contained 200 g/L sucrose, it was observed that if FOS were present in the culture medium, both yeasts exhibited fructosyltransferase activity, producing blastose and 6-kestose. However, *C. lusitaniae* ITMLB85 also produces ^1^F-FOS such as 1-kestose, 1-nystose, and 1-F fructofuranosylnystose. Therefore, fructosyltransferase activity depends on the yeast species.

In the treatments in which inulin was used as a carbon source, both yeasts produced 6-kestose and 1-kestose, and only *P. kudriavzevii* ITMLB97 produced blastose. Similarly, when BS-juice was used as a carbon source, both yeasts produced 6-kestose, and again, only *P. kudriavzevii* ITMLB97 produced blastose. Therefore, in addition to yeast and its fructosyltransferase activity, the presence and concentration of other nutrients and carbohydrates, such as fructose, glucose, sucrose, FOS and HDP-fructans, affect the type of FOS that is produced.

Assays with different carbon sources revealed that the principal fructosyltransferase activities of the yeasts are 1-SST, 6-SST, and 1-FFT. However, an important difference between the use of whole yeast cells and the use of enzyme extracts is that the prebiotic products were generated in shorter time, which is related to an efficient process. However, finding suitable induction media to produce the enzymatic extracts must be achieved.

The evaluation of the fructosyltransferase activity of the different enzymatic extracts of *P. kudriavzevii* ITMLB97 or *C. lusitaniae* ITMLB85 allowed us to determine the optimal conditions for disaccharides and FOS production. However, it is important to emphasize that these optimal values are within the established scope of the study. Therefore, a new approach can be developed based on the results obtained.

According with the results obtained from the HPAEC-PAD chromatograms and their relationship with the SRs generated by the Box–Behnken designs, it was determined that in addition to the sucrose concentration, the glucose concentration also plays a determining role in the production of disaccharides and FOS. However, their formation implies that Ffase (invertase) hydrolyzes sucrose, releasing glucose and fructose into the reaction medium, and performs fructosyltransferase activity, transferring the fructosyl group to sucrose, fructose, or glucose. Therefore, these findings suggest that Ffase exerts invertase and fructosyltransferase activity simultaneously.

*EE-Cl-NM* gathered the most relevant conditions according to its RS for disaccharides and FOS production. Among the products generated, 1-kestose is a prebiotic well established in the food and pharmaceutical industries, and its action at the clinical level has been widely investigated. However, more studies on the potential and function of blastose, 6-kestose, and neo-kestose as prebiotics and therefore their effects on health are needed. Although this study detected disaccharides and FOS, their production at the industrial scale under these conditions is difficult, and further research is needed to obtain high FOS yields and also purified enzymes.

## Figures and Tables

**Figure 1 foods-14-02714-f001:**
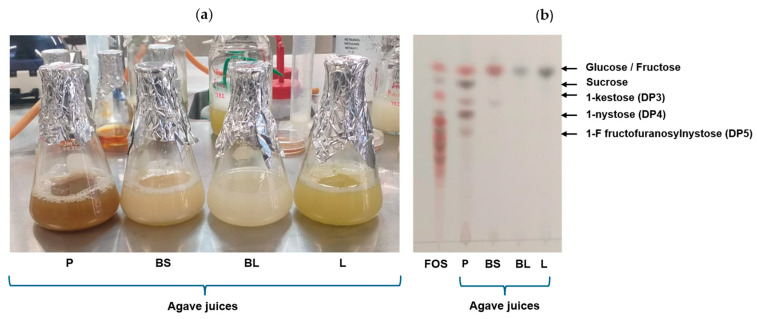
(**a**) Agave juices for fermentation (**P** = pine head juice, **BS** = base of scape juice, **BL** = base of the leaf juice, and **L** = leaf juice); (**b**) TLC of the agave juices using aniline as a derivatizing reagent (**FOS** = fructooligosaccharides standard from Megazyme). The difference in the color of the bands is produced after derivatization with aniline, which depends on the type of glycosidic bond present in the structures. If glucose is present, dark blue bands appear, while if fructose is present, pinkish-reddish bands appear.

**Figure 3 foods-14-02714-f003:**
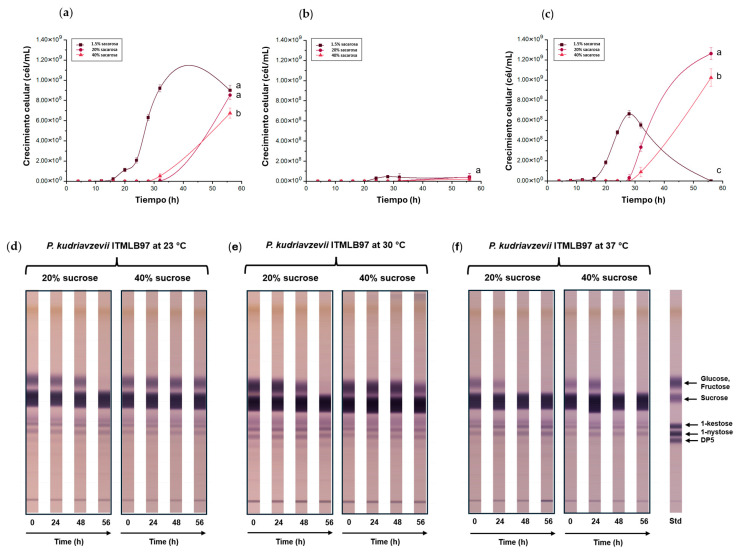
Cell growth comparison of (**a**) *P. kudriavzevii* ITMLB97, (**b**) *K. marxianus* ITMLB106, and (**c**) *C. lusitaniae* ITMLB85 with different sucrose concentrations (1.5, 20, and 40%) at 30 °C. For cell growth at 56 h, different letters indicate significant differences according to Tukey’s test for α = 0.05, *n* = 3. TLC of the fermentation products with (**d**–**f**) *P. kudriavzevii* ITMLB97 and (**g**–**i**) *C. lusitaniae* ITMLB85 at different temperatures (23, 30, and 37 °C), sucrose concentrations (20 and 40%), and times (0, 24, 48, and 56 h). **DP5** = 1-F-fructofuranosylnystose.

**Figure 4 foods-14-02714-f004:**
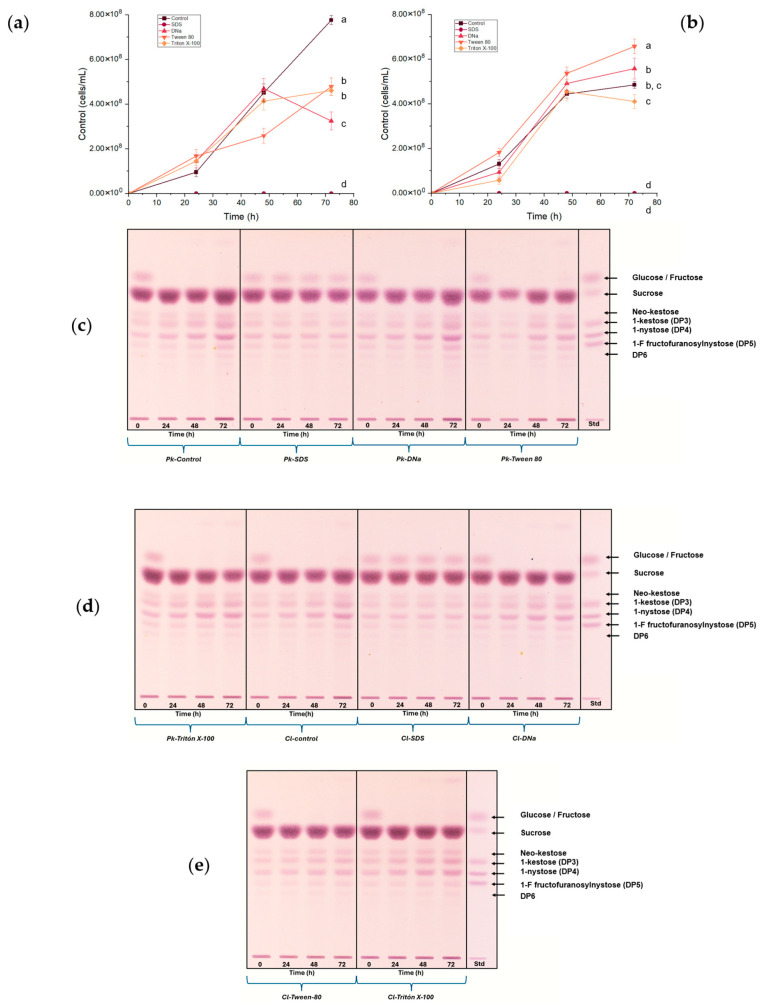
Cell growth of (**a**) *P. kudriavzevii* ITMLB97 and (**b**) *C. lusitaniae* ITMLB85 with surfactants at a 20% sucrose concentration in the media over time. For cell growth at 72 h, different letters indicate significant differences according to Tukey’s test for α = 0.05, *n* = 3. (**c**–**e**) TLC of the fermentation products for both yeasts, where increases in the FOS region over time were observed.

**Figure 5 foods-14-02714-f005:**
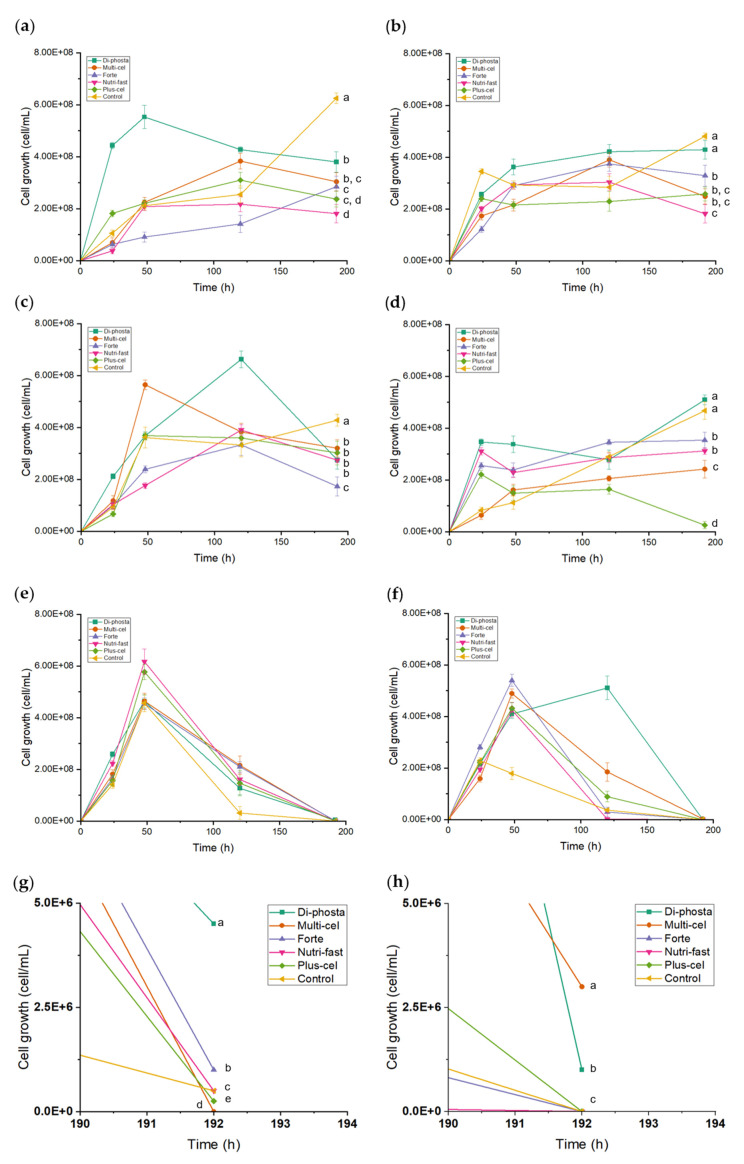
Grown kinetics of (**a**,**c**,**e**) *P. kudriavzevii* ITMLB97 and (**b**,**d**,**f**) *C. lusitaniae* ITMLB85 with different carbon sources and nutrients. *P. kudriavzevii* ITMLB97 used as a carbon source: (**a**) FOS, (**c**) inulin, and (**e**) BS-juice; *C. lusitaniae* ITMLB85 used as a carbon source: (**b**) FOS, (**d**) inulin, and (**f**) BS-juice. (**g**,**h**) Approaches to the kinetics of cell growth of *P. kudriavzevii* ITMLB97 or *C. lusitaniae* ITMLB85 at 192 h, with BS-juice used as a carbon source. For cell growth at 192 h, different letters indicate significant differences according to Tukey’s test for α = 0.05, *n* = 3.

**Figure 6 foods-14-02714-f006:**
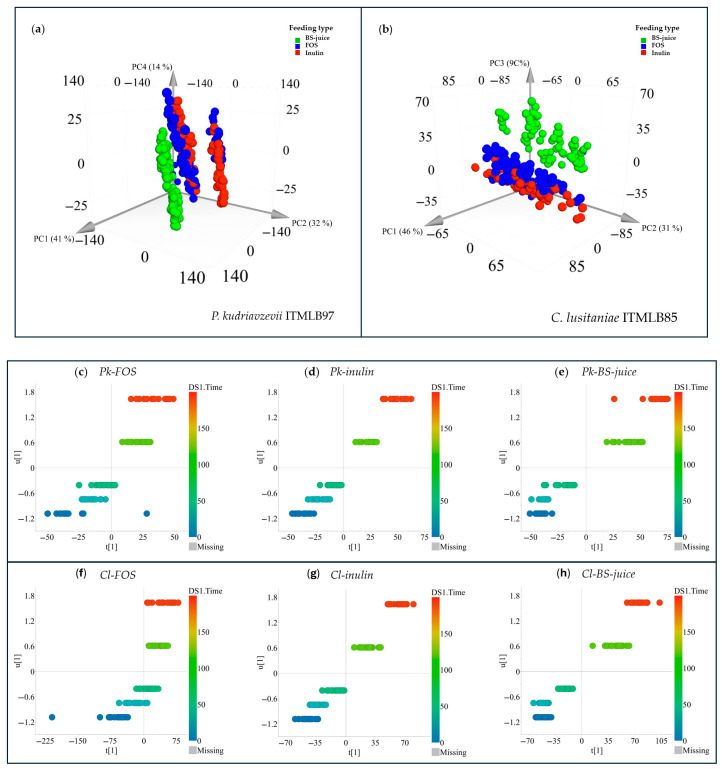
PCA of (**a**) *P. kudriavzevii* ITMLB97 and (**b**) *C. lusitaniae* ITMLB85 with different carbon sources. OPLS for *P. kudriavzevii* ITMLB97 over time using (**c**) FOS, (**d**) inulin, and (**e**) BS-juice. OPLS results for *C. lusitaniae* ITMLB85 over time when (**f**) FOS, (**g**) inulin, and (**h**) BS-juice were used.

**Figure 7 foods-14-02714-f007:**
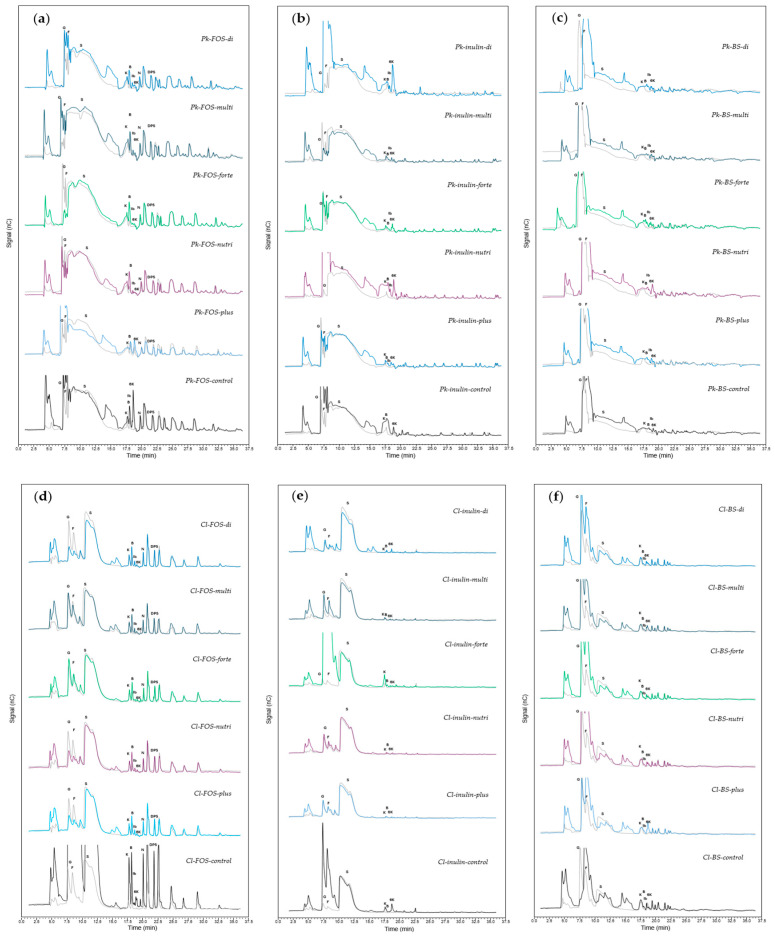
HPAEC-PAD chromatograms of fermentation products from (**a**–**c**) *P. kudriavzevii* ITMLB97 and (**d**–**f**) *C. lusitaniae* ITMLB85 subjected to different carbon sources and nutrients. *P. kudriavzevii* ITMLB97 with different carbon sources: (**a**) FOS, (**b**) inulin, and (**c**) BS-juice; *C. lusitaniae* ITMLB85 with different carbon source: (**d**) FOS, (**e**) inulin, and (**f**) BS-juice. The gray chromatograms correspond to samples at 0 h, and the color chromatograms correspond to treatments at 192 h. **G** = glucose, **F** = fructose, **S** = sucrose, **K** = 1-kestose, **B** = blastose, **Ib** = inulobiose, **6K** = 6-kestose, and **Lb** = levanobiose.

**Figure 8 foods-14-02714-f008:**
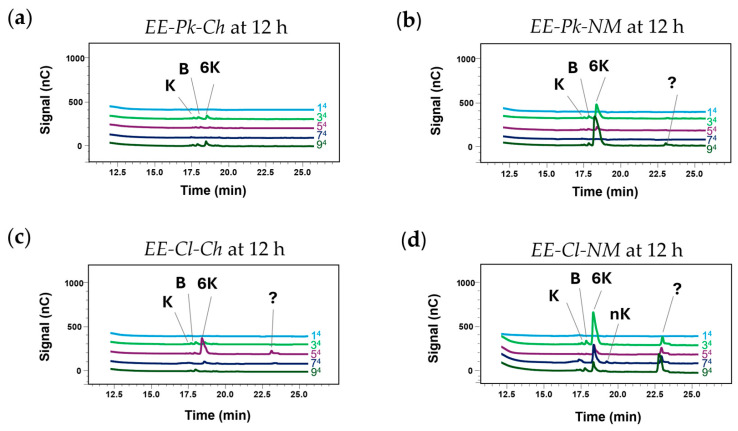
HPAEC-PAD chromatograms of the reaction products from different treatments established by the Box–Behnken designs using distinct enzymatic extracts of *P. kudriavzevii* ITMLB97 or *C. lusitaniae* ITMLB85: (**a**) *EE-Pk-Ch*, (**b**) *EE-Pk-NM*, (**c**) *EE-Cl-Ch*, (**d**) *EE-Cl-NM*. Figures (**a**–**d**) show the products in treatments 1, 3, 5, 7, and 9 at 12 h (indicated by ^4^): **G** = glucose, **F** = fructose, **K** = 1-kestose, B = blastose, **Ib **= inulobiose, **6K** = 6-kestose, **nK** = neo-kestose, and **?** = unknown fructooligosaccharide between DP5-DP6 .

**Figure 9 foods-14-02714-f009:**
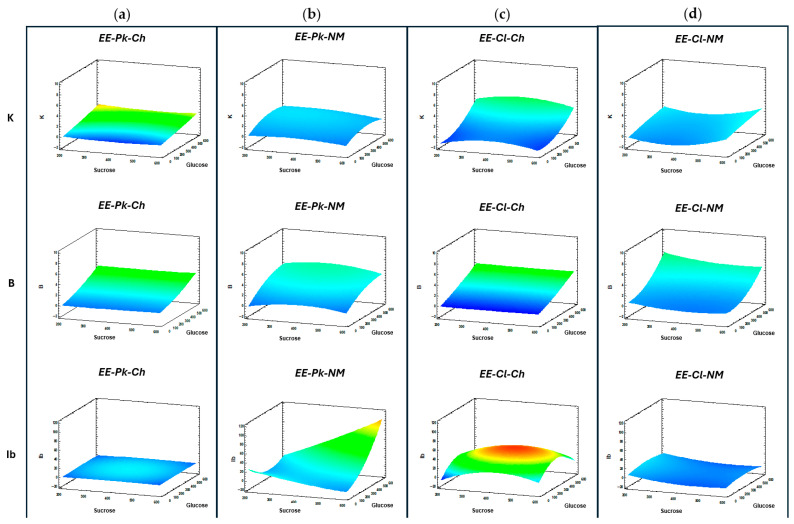
Response surfaces generated for disaccharides and FOS production by enzymatic extracts of *P. kudriavzevii* ITMLB97 (**a**) *EE-Pk-Ch* and (**b**) *EE-Pk-NM* and *C. lusitaniae* ITMLB85 (**c**) *EE-Cl-Ch* and (**d**) *EE-Cl-NM*. **K** = 1-kestose, **B** = blastose, **Ib** = inulobiose, **6K** = 6-kestose, **Lb** = levanobiose, and **nK** = neo-kestose. The increase in disaccharides or FOS production varies depending on the enzyme extract used in the reaction. SR colors are associated with higher or lower disaccharides or FOS production. Production by colors, from lowest to highest, is described below: dark blue, light blue, green, yellow, orange, and red.

**Table 1 foods-14-02714-t001:** Treatments for the evaluation of the effects of carbon sources and nutrients on yeast growth and FOS production.

Carbon Source	Nutrient					
	Di-Phosta	Multicel	Forte	Nutri-Fast	Plus-Cel	Control (Not Nutrient)
**FOS**	*Pk-FOS-di*	*Pk-FOS-multi*	*Pk-FOS-forte*	*Pk-FOS-nutri*	*Pk-FOS-plus*	*Pk-FOS-control*
**inulin**	*Pk-inulin-di*	*Pk-inulin-multi*	*Pk-inulin-forte*	*Pk-inulin-nutri*	*Pk-inulin-plus*	*Pk-inulin-control*
**BS-juice**	*Pk-BS-di*	*Pk-BS-multi*	*Pk-BS-forte*	*Pk-BS-nutri*	*Pk-BS-plus*	*Pk-BS-control*
**FOS**	*Cl-FOS-di*	*Cl-FOS-multi*	*Cl-FOS-forte*	*Cl-FOS-nutri*	*Cl-FOS-plus*	*Cl-FOS-control*
**inulin**	*Cl-inulin-di*	*Cl-inulin-multi*	*Cl-inulin-forte*	*Cl-inulin-nutri*	*Cl-inulin-plus*	*Cl-inulin-control*
**BS-juice**	*Cl-BS-di-*	*Cl-BS-multi*	*Cl-BS-forte*	*Cl-BS-nutri*	*Cl-BS-plus*	*Cl-BS-control*

All the treatments with ***Pk*** were inoculated with *P. kudriavzevii* ITMLB97, whereas all the treatments with ***Cl*** were inoculated with *C. lusitaniae* ITMLB85.

**Table 2 foods-14-02714-t002:** Box–Behnken designs and treatments for the evaluation of the different enzymatic extracts.

	*EE-Pk-Ch*			*EE-Pk-NM*			*EE-Cl-Ch*			*EE-Cl-NM*		
	Glucose (g/L)			Glucose (g/L)			Glucose (g/L)			Glucose (g/L)		
Sucrose (g/L)	0	300	600	0	300	600	0	300	600	0	300	600
**200**	1^Pk-Ch^	2^Pk-Ch^	3^Pk-Ch^	1^Pk-NM^	2^Pk-NM^	3^Pk-NM^	1^Cl-Ch^	2^Cl-Ch^	3^Cl-Ch^	1^Cl-NM^	2^Cl-NM^	3^Cl-NM^
**400**	4^Pk-Ch^	5^Pk-Ch^	6^Pk-Ch^	4^Pk-NM^	5^Pk-NM^	6^Pk-NM^	4^Cl-Ch^	5^Cl-Ch^	6^Cl-Ch^	4^Cl-NM^	5^Cl-NM^	6^Cl-NM^
**600**	7^Pk-Ch^	8^Pk-Ch^	9^Pk-Ch^	7^Pk-NM^	8^Pk-NM^	9^Pk-NM^	7^Cl-Ch^	8^Cl-Ch^	9^Cl-Ch^	7^Cl-NM^	8^Cl-NM^	9^Cl-NM^

The numbers 1, 2, 3, 4, 5, 6, 7, 8, and 9 indicate the treatments established by the different Box–Behnken designs for the corresponding enzymatic extracts (*EE-Pk-Ch*, *EE-Pk-NM*, *EE-Cl-Ch* or *EE-Cl-NM*).

**Table 3 foods-14-02714-t003:** Physicochemical analyses of the agave juices.

Juice Type	pH	Density	°Brix	Moisture %	Ash %	RS (g/L)	Protein (ug/mL)	Total Phenols (ug/mL GAE)
**P**	5.51 ± 0.00 ^a^	1.04± 0.01 ^a^	10.80 ± 0.10 ^a^	92.08 ± 0.20 ^a^	1.67 ± 0.02 ^a^	9.18 ± 0.07 ^a^	41.28 ± 0.05 ^a^	3.8 ± 0.23 ^a^
**BS**	5.44 ± 0.00 ^b^	1.03 ± 0.00 ^a,b^	8.50 ± 0.50 ^b^	96.76 ± 1.25 ^a^	1.24 ± 0.09 ^b,c^	7.35 ± 0.02 ^b^	35.86 ± 0.00 ^b^	3.53 ± 0.18 ^a^
**BL**	5.21 ± 0.00 ^c^	1.02 ± 0.01 ^a,b^	5.40 ± 0.30 ^c^	95.99 ± 0.26 ^a^	1.29 ± 0.02 ^b^	6.97 ± 0.41 ^a,b^	27.87 ± 0.04 ^c^	2.06 ± 0.06 ^b^
**L**	5.02 ± 0.00 ^d^	1.02 ± 0.01 ^b^	4.80 ± 0.10 ^c^	96.70 ± 0.16 ^a^	1.12 ± 0.06 ^c^	5.58 ± 0.18 ^c^	25.74 ± 0.01 ^d^	2.51 ± 0.31 ^b^

**P** = pine head juice, **BS** = base of scape juice, **BL** = base of leaf juice, **L** = leaf juice, **RS** = reducing sugars, and **GAE** = gallic acid equivalents. Different letters indicate significant differences according to Tukey’s test for α = 0.05, *n* = 3.

## Data Availability

The original contributions presented in the study are included in the article/Appendix A, further inquiries can be directed to the corresponding author.

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
