# Peer review of "Disaccharides and Fructooligosaccharides (FOS) Production by Wild Yeasts Isolated from Agave"

_foods, 2025, doi:10.3390/foods14152714_

Round 1

Reviewer 1 Report

Comments and Suggestions for Authors

Review for Belmonte-Izquierdo et al, “Disaccharides and fructooligosaccharides (FOS) production by wild yeasts isolated from Agave” Foods 2025

This article explores the use of agave juice as a substrate for production of FOS by yeast strains. This research is significant due to the potential health benefits of FOS prebiotics. The authors used a locally grown agave plant and extracted different parts of the plant for juice to test as a growth media for various yeast strains. The first half of the results section describes various growth experiments to identify the best-growing yeast and effects of adding sucrose, surfactants, and other nutrient supplements. The authors then characterized the FOS produced in the culture broth and by in vitro reactions using enzymatic extracts from the two best-growing yeast with FTIR, TLC, and HPAEC-PAD. Finally, the authors attempted to optimize conditions for using enzymatic extracts to produce FOS using a Box-Behnken design.

Although there are a few similar published studies on production of FOS from agave (or aguamiel), the main novelty of this study seems to be using different microorganisms for FOS production and agave from a different region of Mexico. Overall, a large amount of data was collected but the text and some figures need significant revision to better explain the rationale for some experimental design and the significance of the results. Please see below for specific comments. 

  • Lines 100-101: If the objective was “to evaluate the fructosyltransferase activity of fifteen yeasts”, the experimental design is not appropriate. To my impression the initial experiments were focused on determining which of the 15 strains could grow on the agave juice, but the fructosyltransferase activity was only evaluated for 3 of the strains. It is possible that if the authors had grown the remaining strains on their induction media, that they would have obtained growth and FOS production from the other strains as well.
  • Lines 102-103: Most of the experiments in this paper revolved around using agave juice as a substrate, but the induction media used to produce the enzymatic extracts and the enzymatic reaction conditions have no relation to agave juice. Please explain the rationale for this seeming change in the focus of the research.
  • Lines 104 & 117-127: Please clarify how these yeast strains were chosen for this study. NRRL strains Y-17537, Y-2231, and Y-1535 do not seem to be isolated from agave (Y-5386 is no longer on their catalog). I was able to find one article mentioning the ITMLB strains were isolated from Mezcal fermentation, but I could not find any details on their isolation or any information for Issatchenkia terricola Y14, Pichia kluyveri Y13, and Kluyveromyces marxianus CDBB-L2029. This is important as the title of the paper includes “… by wild yeasts isolated from Agave”, and this would also help to contextualize the later experiments on enzymatic extracts.
  • Line 107: How many agave specimens were collected? Is there any more information available on what species of agave was/were used? Were these “wild” or cultivated agave plants?
  • Line 130: Please include a manufacturer/model for the refractometer.
  • Line 150: were the different juices all adjusted to this pH? According to table 2, the pH ranged from 5.0 – 5.5.
  • Line 151: According to this, cell counts were recorded every 6 hours, but from section 3.2 and figure 2, it is not mentioned what timepoint is reported for the cell density.
  • Line 166: According to table 2, the Brix of BS juice is approximately 6.5. Please clarify how/when the BS-juice was adjusted to Brix = 8, as different amounts of sucrose were also added. Was it after adding the salts, before adding the sucrose?
  • Line 167: was this the natural pH of the enriched BS-juice or was any acid/base used to adjust the pH to 5.5?
  • Line 174: Was any sucrose added to the BS-juice in this experiment? Also, use the full name for DNa and SDS for the first time they appear.
  • Line 184: This section is missing essential details. What is the source of FOS and inulin? Were the FOS, inulin, and BS-juice added as part of a different formula, and at what concentrations? What volume of medium/flasks were used (It is insufficient to say as described above, since different volumes were used in sections 2.5 and 2.6)?
  • Lines 294-296: This sentence seems to be missing reference(s).
  • Figure 2: How many replicates were used in this experiment?
  • Line 362: This sentence should be moved to the beginning of this section to explain why this experiment was carried out.
  • Figures 3-5: Growth curves should be plotted with a logarithmic y-axis, which can help show differences at lower cell densities without needing a second set of zoomed-in panels.
  • Line 403: Similarly, this sentence should be moved to the beginning of this section to explain why this experiment was carried out.
  • Lines 414-416: I am not sure if figure 4 shows FOS production as clearly as the authors state. Looking at the bands at the bottom of the TLC images, it seems like there were some differences in the amount of sample loaded between bands, which caused all of the bands in certain timepoints to appear more intense. For example, in figure 4C, the bottom bands for 72 h in both Pk-Control and Pk-DNa seem noticeably darker. Do the authors have a way to re-analyze and present this data in a more quantitative method, such as a densitogram?
  • Figure 4b. The letter “a” seems to be missing from the top growth curve.
  • Lines 547-549: The authors make this speculative statement and should be able to examine the available data to test this hypothesis. Perhaps another version of the Figure 6a PCA with data points colored by nutrients could have been added as a supplementary figure.
  • Lines 549-551: In addition to differences between inulin/FOS and the sugars in BS-juice, does BS-juice not also contain various other compounds that could affect the results? Were any FTIR measurements taken for culture media without any yeast inoculation?
  • Lines 611-612: This sentence seems to be referring to Figure 7b, not 7f?
  • Lines 611-647: This seems to be the first time the abbreviation SST is used. Please define.

These paragraphs have multiple speculations on whether a fructofuranosidase (FFase) to fructosyltransferase (SST) is acting and making the observed FOS. With FFase activity, there should also be a decrease in larger FOS, and with SST activity, there should be a decrease in sucrose for any FOS produced. However, it seems that lines 619-620 explain that their HPAEC-PAD methodology is unable to show the larger FOS peaks, and figures 7a-7c do not have clear peaks for sucrose. Therefore, it is hard to evaluate these results and discussions on whether it is FFase or SST responsible for producing the observed FOS.

  • Lines 618-619: I am not sure what relation this sentence has to the above sentences or to which portion of the data in figure 7.
  • Lines 648-680: These paragraphs contain a large amount of discussion on ions in the nutrient formulas, and how they are essential for yeast growth. Firstly, these discussions may be more appropriate following the results presented in figure 5 (growth curves). However, overall, it seems that there is no clear message being made despite a large amount of sentences being written.
  • Figure 8: This figure is very problematic. First, the titles above each panel seem to be in Spanish. Second, each panel is very small and this causes each chromatogram to be so shrunken that it is hard to see some of the peaks. The figure panels should be re-sized, or if possible, the relevant peaks should be integrated and the data could be shown in another way such as with bar graphs, since the methods section seems to say that FOS standard curves were run. Actual quantification of the FOS would make the results in this paper more impactful, since that would allow these results to be compared to methods for producing FOS presented by other papers.
  • Lines 818-848: This paragraph and Table 3 reports FOS production using the units nC. Based on Figure 8 this seems to be the signal strength from the HPAEC-PAD. However, these values provide no information on actual concentrations. As per my previous comment, actual quantification of FOS would be needed to compare the merits of the FOS production method in this paper and other existing methods.
  • Table 3: For Lb produced by EE-Cl-Ch, the optimum time is reported as 0. Does this imply that Lb is present in the starting conditions?
  • Lines 855-856: Figure 1 shows the DP of FOS in the juice of the Agave sp. used in this study, but I do not recall any part of this paper that discussed the DP of FOS in the juice of other Agave?
  • Lines 860-884: Much of this simply restates the results presented above. There is little discussion on any advantages offered by these yeasts or the Agave sp. grown in this region, or what barriers remain for developing this method for large-scale FOS production. 

Author Response

Comment 1) Lines 100-101: If the objective was “to evaluate the fructosyltransferase activity of fifteen yeasts”, the experimental design is not appropriate. To my impression the initial experiments were focused on determining which of the 15 strains could grow on the agave juice, but the fructosyltransferase activity was only evaluated for 3 of the strains. It is possible that if the authors had grown the remaining strains on their induction media, that they would have obtained growth and FOS production from the other strains as well.

Response:

Thank you for your observation. We modified the aim of the study, you are right, and the real aim was to select which strains could grow in the induction media and be used for further experiments. The text was modified on lines 21 and 105.

Comment 2) Lines 102-103: Most of the experiments in this paper revolved around using agave juice as a substrate, but the induction media used to produce the enzymatic extracts and the enzymatic reaction conditions have no relation to agave juice. Please explain the rationale for this seeming change in the focus of the research.

Response:

Thank you for your concern. In fact, the experiments focused on the use of agave juice as a substrate for yeasts in the production of FOS; therefore, to understand, complement and corroborate the type of enzymatic activity that produced the FOS and disaccharides observed, other carbon sources were evaluated, and the enzymatic extracts of the yeasts were obtained, allowing us to corroborate the fructosyltransferase activity of agave yeasts. For the evaluation of fructosyltransferase activity, a modified methodology was proposed on the basis of the conditions used conventionally, in which sucrose is used as the main agent; however, evaluating the effect of glucose concentration on the potential formation of disaccharides and FOS was also desirable; therefore, sucrose and glucose were the two substrates used in the tests with the enzymatic extracts of agave yeasts in the Box–Behnken designs. To address this observation, a correction was made on line 107.

Comment 3) Lines 104 & 117-127: Please clarify how these yeast strains were chosen for this study. NRRL strains Y-17537, Y-2231, and Y-1535 do not seem to be isolated from agave (Y-5386 is no longer on their catalog). I was able to find one article mentioning the ITMLB strains were isolated from Mezcal fermentation, but I could not find any details on their isolation or any information for Issatchenkia terricola Y14, Pichia kluyveri Y13, and Kluyveromyces marxianus CDBB-L2029. This is important as the title of the paper includes “… by wild yeasts isolated from Agave”, and this would also help to contextualize the later experiments on enzymatic extracts.

Response:

The different yeasts available at the Laboratorio de Bioquímica, which belong to 1) the USDA collection and 2) the strain collection of TecNM/Instituto Tecnológico de Cd, were used in this study. Hidalgo, and 3) the strain collection of TecNM/Instituto Tecnológico de Morelia. The fifteen yeasts used in the study were isolated from diverse and different sources. Specifically, within the strain collection of TecNM/Technological Institute of Morelia. In particular, Candida lusitaniae ITMLB85, Kluyveromyces marxianus ITMLB106, and Pichia kudriavzevii ITMLB97 were previously isolated from the working group of Agave mapisaga. The yeasts were characterized via microbiological and molecular techniques, including DNA extraction, PCR amplification with primers ITS1 and ITS4, and restriction fragment length polymorphism (RFLP) analysis, from which the purified products were sequenced and subsequently analyzed via the Basic Local Alignment Search Tool (BLAST) and the National Center for Biotechnology Information (NCBI). To address this observation, corrections were made, the first one on line 133 and the second one on line 184, where three yeasts were selected.

Comment 4) Line 107: How many agave specimens were collected? Is there any more information available on what species of agave was/were used? Were these “wild” or cultivated agave plants?

Response:

We used ten twelve-year-old wild Agave sp. samples collected in Cuitzeo, Michoacán, Mexico. To address this observation, this information was added to the Materials and Methods (line 115).

Comment 5) Line 130: Please include a manufacturer/model for the refractometer.

Thank you for your observation; this information was added to the materials and methods section (line 156).

Comment 6) Line 150: Were the different juices all adjusted to this pH? According to Table 2, the pH ranged from 5.0 – 5.5.

Response:

The juice used for fermentation was adjusted to a pH of 5.5, and hydrochloric acid (HCl) or sodium hydroxide (NaOH) was used to adjust the pH. This information was added to the manuscript (lines 178, 202, 221, 245 and 287).

Comment 7) Line 151: According to this, cell counts were recorded every 6 hours, but from section 3.2 and figure 2, it is not mentioned what timepoint is reported for the cell density.

Response:

We added this information to section 3.2: “At 30 h, any tested yeasts did not grow in BL or L juices (Figure 2)”. In addition, this information was added on line 180 and to the footer of Figure 2, line 490: “Figure 2. Yeast growth in agave juices (P=pine head juice, BS= base of scape juice, BL= base of leaf juice, and L= leaf juice) at 30 h at different temperatures: (a) 25 °C, (b) 35 °C and (c) 45 °C”.

Comment 8) Line 166: According to table 2, the Brix of BS juice is approximately 6.5. Please clarify how/when the BS-juice was adjusted to Brix = 8, as different amounts of sucrose were also added. Was it after adding the salts, before adding the sucrose? 

Response:

Thank you very much for your observation. The value of 6.5 in Table 2 is an error that we did not detect, and it has been corrected in this new version of the manuscript. The °Brix of the agave juice, specifically the BS juice, at the time of the physicochemical analysis was 8.5. This value was important for establishing the °Brix that would be used as a base for fermentation. If any of the BS juices obtained for fermentation had more than 8°Brix, a dilution was made with sterile distilled water before adding the salts and sucrose. To address this point, the correction was made in Table 2, and it was also specified how the Brix degrees were adjusted at lines 191 and 211.

Comment 9) Line 167: was this the natural pH of the enriched BS-juice or was any acid/base used to adjust the pH to 5.5?

Response:

BC juice has a natural pH between 5.4 and 6.3; however, we used 0.1 N hydrochloric acid (HCl) or 0.1 N sodium hydroxide to adjust the pH. To address this point, this information was added to lines 178, 202, 221, 245 and 287.

Comment 10) Line 174: Was any sucrose added to the BS-juice in this experiment?

Response:

The enriched BS juice also contained 200 g/L sucrose in all the treatments. To address this observation, the information was added at lines 186 and 212. 

Comment 11) Also, use the full name for DNa and SDS for the first time they appear.

Response:

Thank you for your observation; the full names were added at line 213.

Comment 12) Line 184: This section is missing essential details. What is the source of FOS and inulin? Were the FOS, inulin, and BS-juice added as part of a different formula, and at what concentrations? What volume of medium/flasks were used (It is insufficient to say as described above, since different volumes were used in sections 2.5 and 2.6)?

Response:

Corrections were made between the proper sections at lines 222-235. Importantly, preinocula of sections 2.5 and 2.6 were obtained via the same procedure in 100 mL Erlenmeyer flasks with 50 mL of YPDE medium. This information was added at line 231. 

Comment 13) Lines 294-296: This sentence seems to be missing reference(s).

Response:

References have been added at line 390. 

Comment 14) Figure 2: How many replicates were used in this experiment?

Response:

Three replicates. To address this observation, the information was added to the figure footnote on line 491 and in the methodology section on line 177.

Comment 15) Line 362: This sentence should be moved to the beginning of this section to explain why this experiment was carried out.

Response:

Thank you for your recommendation; the sentence was moved to the beginning of the section, line 498. 

Comment 16) Figures 3-5: Growth curves should be plotted with a logarithmic y-axis, which can help show differences at lower cell densities without needing a second set of zoomed-in panels.

Response:

To address this observation, new figures (3 to 5) were created and added as supplementary material.

Comment 17) Line 403: Similarly, this sentence should be moved to the beginning of this section to explain why this experiment was carried out.

Response:

The sentence was moved to the beginning of the section, line 551.

Comment 18) Lines 414-416: I am not sure if figure 4 shows FOS production as clearly as the authors state. Looking at the bands at the bottom of the TLC images, it seems like there were some differences in the amount of sample loaded between bands, which caused all of the bands in certain timepoints to appear more intense. For example, in figure 4C, the bottom bands for 72 h in both Pk-Control and Pk-DNa seem noticeably darker. Do the authors have a way to re-analyze and present this data in a more quantitative method, such as a densitogram?

Response:

For TLC analysis, all samples were prepared at the same concentration (7 mg/mL), after which 7 µL of each sample was applied to a silica sheet with a CAMAG Automatic TLC sampler ATS4. Therefore, differences in the intensity of the bands at the application points are attributed to the increase in the degree of polymerization caused by each treatment, which is associated with the fructosyltransferase activity of the yeast and not with the different amounts of sample applied to the sheet. To address this comment, the methodology is described on line 326.

Comment 19) Figure 4b. The letter “a” seems to be missing from the top growth curve.

Response:

Figure 4 has been corrected.

Comment 20) Lines 547-549: The authors make this speculative statement and should be able to examine the available data to test this hypothesis. Perhaps another version of the Figure 6a PCA with data points colored by nutrients could have been added as a supplementary figure.

Response:

English: Thank you for your recommendations. The figure was generated and added to the supplementary figures.

Comment 21) Lines 549-551: In addition to differences between inulin/FOS and the sugars in BS-juice, does BS-juice not also contain various other compounds that could affect the results? Were any FTIR measurements taken for culture media without any yeast inoculation?

Response:

The analyses performed via TLC and HPAEC-PAD revealed that BS juice contains simple carbohydrates such as G, F, and S and different FOS. These products change during fermentation and are monitored over time. Furthermore, the bands detected by FT-IR are associated with the carbohydrate backbone, whereas the other bands are specifically associated with fructans or fructose. Therefore, the presence or effect of any other compound that could affect the results is ruled out. On the other hand, the FT-IR data without yeast inoculation were identical to those of the samples taken at the initial time point of each treatment.

Comment 22) Lines 611-612: This sentence seems to be referring to Figure 7b, not 7f?

Response:

Corrections were made to ensure a proper match between the text and the figure.

Comment 23) Lines 611-647: This seems to be the first time the abbreviation SST is used. Please define.

Response:

The abbreviations for enzymes 1-SST, 6-SST and 1-FFT are defined at line 802.

Comment 24) These paragraphs have multiple speculations on whether a fructofuranosidase (FFase) to fructosyltransferase (SST) is acting and making the observed FOS. With FFase activity, there should also be a decrease in larger FOS, and with SST activity, there should be a decrease in sucrose for any FOS produced. However, it seems that lines 619-620 explain that their HPAEC-PAD methodology is unable to show the larger FOS peaks, and figures 7a-7c do not have clear peaks for sucrose. Therefore, it is hard to evaluate these results and discussions on whether it is FFase or SST responsible for producing the observed FOS.

Response:

Thank you for your observations.

Comment 24 A. These paragraphs have multiple speculations on whether a fructofuranosidase (FFase) to fructosyltransferase (SST) is acting and making the observed FOS. 

Response:

Yeasts contain β-fructofuranosidase enzymes (invertases, FFases), which exhibit fructosyltransferase activity at high sucrose concentrations. Enzymes with fructosyltransferase (FTase) activity are classified on the basis of the product that they produce:

  • 1-SST (sucrose:sucrose 1-fructosyltransferase), which produces 1-kestose
  • 6-SST (sucrose:sucrose 6-fructosyltransferase), which produces 6-kestose
  • 1-FFT (fructan:fructan 1-fructosyltransferase), which produces 1F-FOS (such as 1-nystose, fructosylnystose, etc.)
  • 6G-FFT (fructan:fructan 6G-fructosyltransferase), which produces neokestose

Therefore, depending on the observed product, the type of fructosyltransferase activity can be deduced/suggested.

Comment 24 B. With FFase activity, there should also be a decrease in larger FOS, and with SST activity, there should be a decrease in sucrose for any FOS produced. 

Response:

To produce FOS via "conventional" fructosyltransferase activity, sucrose is expected to accept the fructosyl group from another sucrose, effectively resulting in a decrease in sucrose; however, recent studies have shown that for the formation of disaccharides such as blastose, inulobiose and levanobiose, glucose and fructose can act as fructosyl group acceptors, provided that high concentrations of sucrose are present in the medium. In some cases, in this study, no decrease in sucrose content was observed. However, no decrease in high-polymerization fructans is observed either, which shows that the products formed are not due to inulinases (hydrolases); in addition, inulinases mainly form fructans of the F series. Therefore, Figures 7b and 7e allow us to rule out inulinase activity, strengthening the hypothesis of fructosyltransferase activity, in which FOSs are produced from small molecules such as sucrose, glucose, and fructose.

Comment 24 C. However, it seems that lines 619-620 explain that their HPAEC-PAD methodology is unable to show the larger FOS peaks, and figures 7a-7c do not have clear peaks for sucrose. Therefore, it is hard to evaluate these results and discussions on whether it is FFase or SST responsible for producing the observed FOS.

Response:

 Thanks for your observation; there was a writing mistake in the text between lines 619--620: “The accumulation of those carbohydrates could be attributed to the hydrolysis of higher-DP fructans, but this was not observed in the chromatograms”. Our intention was not to indicate that the HPAEC-PAD methodology cannot show the larger FOS peaks because FOS with a DP ≥ 5 are clearly observed in Figures 7a, 7b, 7d and 7e, between retention times of 20 and 37.5 min. The text attempted to discuss the idea that the blastose, 1-kestose, and 6-kestose, among others, are not the result of inulinases (hydrolases) but de novo synthetized molecules, which are not reduced in the intensity of high-polymerization fructans. To clarify this idea, the text was modified in the corresponding lines.

Comment 25) Lines 618-619: I am not sure what relation this sentence has to the above sentences or to which portion of the data in figure 7.

Response:

The lines were rewritten to make them clearer.

Comment 26) Lines 648-680: These paragraphs contain a large amount of discussion on ions in the nutrient formulas, and how they are essential for yeast growth. Firstly, these discussions may be more appropriate following the results presented in figure 5 (growth curves). However, overall, it seems that there is no clear message being made despite a large amount of sentences being written.

Response:

We rewrote the paragraph to make it more concise and clearer. The discussion for Figures 5 and 6 was rewritten (lines 630 and 784).

Comment 27) Figure 8: This figure is very problematic. First, the titles above each panel seem to be in Spanish. Second, each panel is very small and this causes each chromatogram to be so shrunken that it is hard to see some of the peaks. The figure panels should be re-sized, or if possible, the relevant peaks should be integrated and the data could be shown in another way such as with bar graphs, since the methods section seems to say that FOS standard curves were run. Actual quantification of the FOS would make the results in this paper more impactful, since that would allow these results to be compared to methods for producing FOS presented by other papers.

Response:

To address this observation, Figure 8 was modified to focus on the 12 h treatments; the titles were corrected, and the figures were enlarged (line 924). Furthermore, the explanation and discussion were clarified on lines 874. Although pure standards for glucose, fructose, sucrose, 1-kestose, 1-nystose, and fructosylnystose are available, in the case of blastose, 6-kestose, neokestose, inulobiose, and levanobiose, pure standards are not available but rather a mixture of them (donated). Therefore, we do not have a calibration curve for the latter. However, these mixtures have allowed us to identify different types of FOS: 1F-FOS, 6F-FOS, neo-FOS, and the base of blasto-FOS.

Comment 28) Lines 818-848: This paragraph and Table 3 reports FOS production using the units nC. Based on Figure 8 this seems to be the signal strength from the HPAEC-PAD. However, these values provide no information on actual concentrations. As per my previous comment, actual quantification of FOS would be needed to compare the merits of the FOS production method in this paper and other existing methods.

Response:

First, the identification of disaccharides and FOS has been quite complex. First, the evaluation initially focused on conventional FOS (1-kestose, 1-nystose, and fructosylnystose), for which we have standards in the working group. However, when the samples were analyzed, we observed that there were more peaks in the HPAEC-PAD chromatograms that did not match the standards. HPAEC-PAD allows the separation of carbohydrates with different degrees of polymerization and allows the separation of isomers, which, because they have different bonds in their structure, are detected as distinct peaks in the chromatogram. This led us to analyze the samples in more detail and look for a way to identify them. Owing to the donation of other working groups (D.C. Clarita Olvera Carranza for standard blastosis and D.C. Norio Shiomi and Midori Yoshida for standard DP´3s), we obtained small volumes of FOS mixtures: blastoses (blastose, levanobiose) and DP´3s (1-kestose, 6-kestose and neokestose), which were essential for naming the disaccharides and FOS found. However, the lack of pure standards of blastose, inulobiose, levanobiose, 6-kestose, and neokestose has limited their quantification. Therefore, we used the detection signal (nC) for each of these compounds to correlate them with their production, as it is known that an increase in the signal is associated with a greater amount of the product of interest. However, we understand the limitation of using the signal (nC) and always report it in this way since calibration curves are unavailable. However, we believe that the identification of these disaccharides and FOS from yeast fermentation provides relevant information on their biotechnological potential. Undoubtedly, new works focused on quantification and comparison should be carried out. To address your comment, we have eliminated Table 3 and detailed the use of the detection signal in the methodology and results sections at lines 291 and 929.

Comment 29) Table 3: For Lb produced by EE-Cl-Ch, the optimum time is reported as 0. Does this imply that Lb is present in the starting conditions?

Response:

Initially, there was no levanobiose in the reaction tubes; the only substrates were sucrose and glucose (at the concentration established by the Box–Behnken design). On the basis of the experimental data, the Box–Behnken design was fed, and it generated an optimization. However, although the design is a valuable response surface model (RSM) tool that contributes to the understanding of the effects of various factors, it also has limitations because it does not have all the data of the complete design space (representation of the area covered by a set of independent factors), resulting in estimated models that could be corroborated. Therefore, this result suggests that the actual optimal value is very close to t = 0 h, indicating that levanobiose formation is a nearly instantaneous reaction. However, this value could be corroborated in the future by a design focused on levanobiose production over time.

Comment 30) Lines 855-856: Figure 1 shows the DP of FOS in the juice of the Agave sp. used in this study, but I do not recall any part of this paper that discussed the DP of FOS in the juice of other Agave?

Response:

To address this point, DP-fructan variation in Agaves of different ages in lines 409 and 975 is discussed.

Comment 31) Lines 860-884: Much of this simply restates the results presented above. There is little discussion on any advantages offered by these yeasts or the Agave sp. grown in this region, or what barriers remain for developing this method for large-scale FOS production.

Response:

To address this observation, the entire conclusions section was rewritten (line 977).

Reviewer 2 Report

Comments and Suggestions for Authors

The manuscript addresses a highly relevant and promising topic, the utilization of agave juices and yeast fructosyltransferases for fructooligosaccharide (FOS) production, with potential applications in the functional food industry.

Here i report some comments that should be addressed:

-The terminology for fructooligosaccharides (FOS) is inconsistent and potentially confusing (lines 41-42). Clearly differentiate the definitions for FOS (DP 3-12) and HDP-fructans (DP>12).

- Lines 43-50: The discussion of enzymes (FTase and FFase) and their activities appears somewhat disconnected from the preceding classification of fructans.

- The rationale for focusing specifically on agave (particularly from Lake Cuitzeo) is underdeveloped. Explain clearly why agave species from this geographic region merit specific investigation beyond general agave species.

- The exact source and history of yeast strains are clearly stated, but the rationale behind selecting these specific strains is absent.
Clarify briefly why these strains were chosen and whether they have a known fructosyltransferase capacity.

- Using a Neubauer chamber alone may introduce inaccuracies or inconsistencies; OD measurements or viable count plating would strengthen reliability.
Provide a rationale for choosing this method and acknowledge its limitations.

- The selected sucrose concentrations (1.5%, 20%, and 40%) appear arbitrary or not clearly justified.

- The rationale for choosing the specific surfactants (DNa, SDS, Tween 80, Triton X-100) and their concentration (10 mM) must be justified clearly by referencing previous studies or preliminary tests.

- Lines 275-277: Stating “notorious physicochemical differences” is vague

- Table 2: Statistical differences indicated by letters are not clearly explained in the text. Explicitly state what specific differences mean biologically or practically (e.g., why is the protein content variation significant?).

- Lines 333-338: The hypothesis about antimicrobial metabolites in BL and L juices is speculative. Support this with references.

- Lines 419-424: Mentioning surfactants' permeabilization effects lacks depth.
Provide specific references or biochemical explanations as to how these surfactants could facilitate substrate incorporation or metabolite export.

-Lines 532-596: While PCA and OPLS models are statistically robust, their biological interpretation is vague... clearly state what these changes mean biologically regarding yeast metabolism, fructan utilization, and FOS production.

- Lines 725-727: Clearly define what exactly "enzymatic extracts" (EE-Pk-Ch, EE-Pk-NM, EE-Cl-Ch, EE-Cl-NM) represent biochemically. Is it crude extract, partially purified, or something else?

- Lines 761-774: Descriptions like "produced at 0 h" are scientifically problematic, as enzymatic production "at 0 h" implies no enzymatic reaction has occurred.

- Include explicit quantitative data (chromatographic peak integration, yields, enzyme activity units) throughout to substantiate claims clearly.

Author Response

Dear reviewer, thank you very much for all your comments; they have been essential for enriching and improving the work. Below are the improvements made to fix it.

Comment 1) The terminology for fructooligosaccharides (FOS) is inconsistent and potentially confusing (lines 41-42). Clearly differentiate the definitions for FOS (DP 3-12) and HDP-fructans (DP>12).

Response:

To address this observation, we modified the text to make the definitions of FOS and HDP-fructans clearer on line 43.

Comment 2) Lines 43-50: The discussion of enzymes (FTase and FFase) and their activities appears somewhat disconnected from the preceding classification of fructans.

Response:

To address this observation, we reorganized the information between lines 44-54 and then delved into the enzymatic activity of FTase and FFase.

Comment 3) The rationale for focusing specifically on agave (particularly from Lake Cuitzeo) is underdeveloped. Explain clearly why agave species from this geographic region merit specific investigation beyond general agave species.

Response:

To address this observation, we expanded the discussion on line 93.

Comment 4) The exact source and history of yeast strains are clearly stated, but the rationale behind selecting these specific strains is absent. Clarify briefly why these strains were chosen and whether they have a known fructosyltransferase capacity.

Response:

To address this observation, corrections were made, the first one at line 133 and the second one at line 184, where it is described why these three yeasts were selected. Furthermore, it should be mentioned that none of the yeasts used in this work had been used to explore their fructosyltransferase activity.

Comment 5) Using a Neubauer chamber alone may introduce inaccuracies or inconsistencies; OD measurements or viable count plating would strengthen reliability.
Provide a rationale for choosing this method and acknowledge its limitations.

Response:

Compared with other methods, the Neubauer chamber is a technique with its own limitations, such as optical density (OD). For example, it is slow, can present variability in counting owing to human error, and requires dilutions if the cell density is very high. Its complementation with other techniques, such as OD, would undoubtedly be very useful. However, for the purpose of this research, the Neubauer chamber allowed real monitoring of cell morphology throughout fermentation, allowing us to observe changes in cell size, shape, and color and therefore determine whether the yeast has experienced any stress, is in optimal condition, or has died. DO measurements involve both live and dead cells. In addition, replicates were performed during fermentation, and the counting protocol was strictly followed.

Comment 6)  The selected sucrose concentrations (1.5%, 20%, and 40%) appear arbitrary or not clearly justified.Response:

The yeasts used throughout the study had not been studied or evaluated for fructosyltransferase activity. On the basis of the literature, we found that yeasts contain fructofuranosidase enzymes (invertases), which commonly hydrolyze sucrose; however, at high sucrose concentrations, these enzymes exhibit fructosyltransferase activity. Among these reports, we reported some very interesting results, in which fructofuranosidase exhibited hydrolytic activity at 5% sucrose and that increasing the sucrose concentration above 10% also increased fructosyltransferase activity (Gomes-Barbosa et al., 2018). Additionally, other studies have used 60% sucrose concentrations to evaluate this activity (Alvaro-Benito et al., 2007). Therefore, we defined a broad range close to these values (1.5, 20, and 40% sucrose).

Comment 7) The rationale for choosing the specific surfactants (DNa, SDS, Tween 80, Triton X-100) and their concentration (10 mM) must be justified clearly by referencing previous studies or preliminary tests.

Response:

To address this observation, this information was added on lines 208 and 551.

Comment 8)  Lines 275-277: Stating “notorious physicochemical differences” is vague

Response:

To address this observation, a modification was made to the text on line 363.

Comment 9) Table 2: Statistical differences indicated by letters are not clearly explained in the text. Explicitly state what specific differences mean biologically or practically (e.g., why is the protein content variation significant?).

Response:

To address this question, information was added to the methodology and information on the significant differences in the physicochemical analyses at lines 159, 204, 224, 247 and 363.

Comment 10) Lines 333-338: The hypothesis about antimicrobial metabolites in BL and L juices is speculative. Support this with references.

Response:

To address this question, information was added at line 434.

Comment 11) Lines 419-424: Mentioning surfactants' permeabilization effects lacks depth.
Provide specific references or biochemical explanations as to how these surfactants could facilitate substrate incorporation or metabolite export.

Response:

To address this question, information was added at line 570.

Comment 12) Lines 532-596: While PCA and OPLS models are statistically robust, their biological interpretation is vague... clearly state what these changes mean biologically regarding yeast metabolism, fructan utilization, and FOS production.

Response: 

Thank you for your suggestion. We increased the explanation for the PCA results and added specific analyses to check nutrient formulation effects, specifically the SIMCA analysis.

Comment 13)  Lines 725-727: Clearly define what exactly "enzymatic extracts" (EE-Pk-Ch, EE-Pk-NM, EE-Cl-Ch, EE-Cl-NM) represent biochemically. Is it crude extract, partially purified, or something else?

Response:

To address this observation, the information is detailed in methodology section 2.8, line 274. 

Comment 14) Lines 761-774: Descriptions like "produced at 0 h" are scientifically problematic, as enzymatic production "at 0 h" implies no enzymatic reaction has occurred.

Response:

Thank you very much for your comment. The text was poorly written. The original intention was to refer to the chromatograms obtained from different treatments at 0 and 24 h of reaction time; however, this figure has finally been removed.

Comment 15) Include explicit quantitative data (chromatographic peak integration, yields, enzyme activity units) throughout to substantiate claims clearly.

Response:

Thank you for your comment. First, the identification of disaccharides and FOS has been quite complex. First, the evaluation initially focused on conventional FOS (1-kestose, 1-nystose, and fructosylnystose), for which we have standards in the working group. However, when the samples were analyzed, we observed that there were more peaks in the HPAEC-PAD chromatograms that did not match the standards. HPAEC-PAD allows the separation of carbohydrates with different degrees of polymerization and allows the separation of isomers, which, because they have different bonds in their structure, are detected as distinct peaks in the chromatogram. This led us to analyze the samples in more detail and look for a way to identify them. Owing to the donation of other working groups (D.C. Clarita Olvera Carranza for standard blastose and D.C. Norio Shiomi and Midori Yoshida for standard DP´3s), we obtained small volumes of FOS mixtures: blastoses (blastose, levanobiose) and DP´3s (1-kestose, 6-kestose and neokestose), which were essential for naming the disaccharides and FOS found. However, the lack of pure standards of blastose, inulobiose, levanobiose, 6-kestose, and neokestose has limited their quantification. Therefore, we used the detection signal (nC) for each of these compounds to correlate them with their production, as it is known that an increase in the signal is associated with a greater amount of the product of interest. However, we understand the limitation of using the signal (nC) and always report it in this way since calibration curves are unavailable. However, we believe that the identification of these disaccharides and FOS from yeast fermentation provides relevant information on their biotechnological potential. Undoubtedly, new works focused on quantification, yields and enzyme activity units should be carried out. To address your comment, we have detailed the use of the detection signal in the methodology and results sections at lines 291 and 929.

Reviewer 3 Report

Comments and Suggestions for Authors

Introduction:

  • The sentences are long and overloaded with information, which hinders readability and comprehension (e.g., the sentence from lines 62 to 65).

  • The term “FFase” may suggest a typographical error.

  • The phrase “for this serie” (line 59) is a clear mistake; it should be “for this series.”

Some sections are oversaturated with references, which distracts from the core content (e.g., a single phrase containing references like [3M], [5], [6–8]). Excessive citation frequency may suggest a compilation rather than a critically analyzed text.Methods and materials

Lack of precise information regarding the juice extraction methodology, such as the extraction conditions: time, force, reproducibility of the procedure, or temperature control during extraction.
The sterilization and storage of samples were addressed too superficially: it was not indicated whether the loss of bioactive compounds (e.g., FOS or polyphenols) was monitored, which could influence subsequent fermentation.
It was not specified whether purity tests or molecular identification (e.g., PCR or sequencing) of the yeast strains were performed prior to use, which is currently a standard practice in studies of this type.
The volumes and inoculation methods were provided, but without reporting specific measurement results (e.g., number of cells in the inoculum, standard deviations, number of replicates). In its current form, it is impossible to assess the statistical reliability of the obtained data.
Lack of justification for the choice of yeast strains and temperature range:
Why was this particular set of yeasts selected? Do they have industrial, biotechnological, or merely local significance? Similarly, no rationale was given for the choice of the tested temperatures (25, 35, and 45 °C). Do these reflect actual fermentation conditions in industry?

Overly General Characterization of the Fermentation Process. Methodological Gaps in the Evaluation of Enzymatic Activity (Section 2.8)

The assessment of fructosyltransferase activity lacks details regarding the validation of analytical methods (TLC, HPAEC-PAD). There is also no information on the repeatability of measurements or the calibration of equipment. The high substrate concentrations used (up to 600 g/L sucrose) pose significant challenges in terms of viscosity and could potentially affect enzyme activity—yet this aspect was not addressed or discussed.

There is a lack of consistency in the volumes of media and flasks used—some experiments were conducted in 250 mL flasks with 100 mL of medium, while others used 50 mL flasks with 30 mL of medium. These differences can influence aeration and the kinetics of yeast growth. However, such variations were neither explained nor controlled for.

The results and conclusions presented in the study are primarily descriptive in nature and, in many instances, merely paraphrase the data without an adequate level of synthesis or deeper scientific reflection. The authors did not attempt to generalize the findings or relate them to a broader biotechnological, industrial, or ecological context.

The emphasis on the unusual degree of polymerization (DP) of agave juices from the Cuitzeo Lake region is interesting, but it would require more compelling justification, preferably supported by comparisons with other species and habitats. The hypothesis regarding the influence of soil conductivity on the chemical composition of the juices remains speculative—as it is not supported by any soil physicochemical analysis results or statistical correlations.

Although the study suggests that BS-juice is a good substrate for yeast cultivation, no concrete practical applications have been proposed—such as in the production of prebiotics, bioethanol, or for industrial fermentation. Moreover, there is a lack of information on the economic feasibility or potential limitations of using this medium.

The application of the Box–Behnken design allowed for the determination of optimal synthesis conditions for certain fructans. However, the conclusions presented are reduced to a mere listing of maximum values without any interpretation of their biological, technological, or practical significance.

At the end of the section, the potential of producing low-DP fructans (DP 3–6) as a source of prebiotics is mentioned, which appears to be the main achievement of the study. Unfortunately, this important conclusion is not adequately developed or linked to the existing literature on the prebiotic activity of such compounds.

Author Response

Dear reviewer, thank you very much for all your comments; they have been essential for enriching and improving the work. Below are the improvements made to fix it.

Comment 1) The sentences are long and overloaded with information, which hinders readability and comprehension (e.g., the sentence from lines 62 to 65).

Response:

Thank you for your observation. To address this observation, the text was rewritten to make it more understandable on line 62.

Comment 2) The term “FFase” may suggest a typographical error.

Response:

For your observation, nonetheless, the abbreviation is correct; we used it to refer to β-fructofuranosidase (FFase).

Comment 3) The phrase “for this serie” (line 59) is a clear mistake; it should be “for this series.”

Response:

This mistake was corrected as suggested.

Comment 4) Some sections are oversaturated with references, which distracts from the core content (e.g., a single phrase containing references like [3M], [5], [6–8]). Excessive citation frequency may suggest a compilation rather than a critically analyzed text.

Response:

To address this observation, we removed the brackets around the molar concentration and moved some references to the end of the sentences so as not to distract from the main idea.

Comment 5) Lack of precise information regarding the juice extraction methodology, such as the extraction conditions: time, force, reproducibility of the procedure, or temperature control during extraction.

Response:

To address this observation, we added information at line 118.

Comment 6) The sterilization and storage of samples were addressed too superficially: it was not indicated whether the loss of bioactive compounds (e.g., FOS or polyphenols) was monitored, which could influence subsequent fermentation.

Response:

To address this observation, important information regarding the sterilization and storage of samples was added at line 124. Importantly, owing to interest in carbohydrates, particularly fructans, in agave juice, TLC was performed to determine the fructan composition of agave juice before and after heat sterilization. As a result, we observed no variation; therefore, we chose heat sterilization as the method to preserve the agave juice.

Comment 7) It was not specified whether purity tests or molecular identification (e.g., PCR or sequencing) of the yeast strains were performed prior to use, which is currently a standard practice in studies of this type.

Response:

This study used different yeasts available at the Laboratorio de Bioquímica, which belong to 1) the USDA collection and 2) the strain collection of TecNM/Instituto Tecnológico de Cd. Hidalgo, and 3) the strain collection of TecNM/Instituto Tecnológico de Morelia. The fifteen yeasts used in the study were isolated from diverse and different sources. Specifically, within the strain collection of TecNM/Technological Institute of Morelia, Candida lusitaniae ITMLB85, Kluyveromyces marxianus ITMLB106, and Pichia kudriavzevii ITMLB97, these three strains were previously isolated from the working group of Agave mapisaga and were characterized via microbiological and molecular techniques, including DNA extraction, PCR amplification with primers ITS1 and ITS4, and restriction fragment length polymorphism (RFLP) analysis, from which the purified products were sequenced and subsequently analyzed via the Basic Local Alignment Search Tool (BLAST) and the National Center for Biotechnology Information (NCBI). To address this observation, corrections were made, the first one on line 133 and the second one on line 184, where it is described why these three yeasts were selected.

Comment 8) The volumes and inoculation methods were provided, but without reporting specific measurement results (e.g., number of cells in the inoculum, standard deviations, number of replicates). In its current form, it is impossible to assess the statistical reliability of the obtained data.

Response:

Thank you very much for your comment. To address this observation, this information was added to the methodological section at lines 155, 177, 203, 222, 230, 246 and 303.

Comment 9) Lack of justification for the choice of yeast strains and temperature range:
Why was this particular set of yeasts selected? Do they have industrial, biotechnological, or merely local significance? Similarly, no rationale was given for the choice of the tested temperatures (25, 35, and 45 °C). Do these reflect actual fermentation conditions in industry?

Response:

The evaluation of yeast growth in different agave juices revealed that the juice where they grew most abundantly was BS juice. Furthermore, of the fifteen yeasts evaluated, only five grew in the juices (C. lusitaniae ITMLB85, C. lusitaniae ITMLB103, K. marxianus CDBB-L2029, K. marxianus ITMLB106, and P. kudriavzevii ITMLB97), three of which were previously isolated from Agave mapisaga and molecularly characterized by the working group (K. marxianus ITMLB106, P. kudriavzevii ITMLB97 and C. lusitaniae ITMLB85), and they also belong to different genera, which is why they were selected for further studies. This information was added at lines 145 and 184. The original temperature range of 25, 35 and 45 °C was selected because optimal growth temperatures for these yeast species have been reported within this range. 

Comment 10) Overly General Characterization of the Fermentation Process. Methodological Gaps in the Evaluation of Enzymatic Activity (Section 2.8)

Response:

To address the observation, information was added at line 256.

Comment 11) The assessment of fructosyltransferase activity lacks details regarding the validation of analytical methods (TLC, HPAEC-PAD). There is also no information on the repeatability of measurements or the calibration of equipment. The high substrate concentrations used (up to 600 g/L sucrose) pose significant challenges in terms of viscosity and could potentially affect enzyme activity—yet this aspect was not addressed or discussed.

Response:

To address this observation, the TLC and HPAEC-PAD methods were performed on lines 326 and 346, respectively. Indeed, high sucrose concentrations, ranging from 200 to 600 g/L, were challenging for both the TLC and HPAEC-PAD methods. However, after several tests, the ideal concentration for observing the FOS region in the chromatograms and maintaining the HPAEC-PAD pressure within the established range was determined. This was also achieved with the help of Milli-Q water injections interspersed between each sample. However, this doubled the analysis time, and each run lasted 75 min.

Comment 12) There is a lack of consistency in the volumes of media and flasks used—some experiments were conducted in 250 mL flasks with 100 mL of medium, while others used 50 mL flasks with 30 mL of medium. These differences can influence aeration and the kinetics of yeast growth. However, such variations were neither explained nor controlled for.

Response:

Thank you for your feedback. The 250 mL flasks with 100 mL of medium were used to obtain all preinocula throughout the study, as well as the induction media, mainly because it is a volume in which we know the optimal number of microbiological batches to inoculate, as well as the ideal incubation time to have viable yeast cells (P. kudriavzevii ITMLB97 and C. lusitaniae ITMLB85) that allow quality preinocula to be obtained. Furthermore, this volume was used in the fermentations to determine the effect of sucrose concentration on FOS production; however, the required volume of BS juice (100 mL) was a limitation, since it would require a large amount of raw material. In addition, the number of flasks that could be placed in the incubator was limited. Therefore, 50 mL flasks with 30 mL of medium were used for fermentation. However, it is important to highlight that within the same treatment, flasks of different volumes were not used to avoid adding factors within the same experiment.

Comment 13) The results and conclusions presented in the study are primarily descriptive in nature and, in many instances, merely paraphrase the data without an adequate level of synthesis or deeper scientific reflection. The authors did not attempt to generalize the findings or relate them to a broader biotechnological, industrial, or ecological context.

Response:

To address this observation, the results and the entire conclusions section were rewritten.

Comment 14) The emphasis on the unusual degree of polymerization (DP) of agave juices from the Cuitzeo Lake region is interesting, but it would require more compelling justification, preferably supported by comparisons with other species and habitats. The hypothesis regarding the influence of soil conductivity on the chemical composition of the juices remains speculative—as it is not supported by any soil physicochemical analysis results or statistical correlations.

Response:

To address this point, the variation in DP-fructan in other agave species of different ages and from other geographic regions is discussed on line 409. Furthermore, the speculative hypothesis regarding the influence of soil conductivity on the agave DP was eliminated (line 975).

Comment 15) Although the study suggests that BS-juice is a good substrate for yeast cultivation, no concrete practical applications have been proposed—such as in the production of prebiotics, bioethanol, or for industrial fermentation. Moreover, there is a lack of information on the economic feasibility or potential limitations of using this medium.

Response:

To address this point, information was added to the conclusions as perspectives.

Comment 16) The application of the Box–Behnken design allowed for the determination of optimal synthesis conditions for certain fructans. However, the conclusions presented are reduced to a mere listing of maximum values without any interpretation of their biological, technological, or practical significance.

Response:

To address this comment, the conclusion focused on Box–Behnken designs was summarized and restated, and the above can be observed at line 1009.

Comment 17) At the end of the section, the potential of producing low-DP fructans (DP 3–6) as a source of prebiotics is mentioned, which appears to be the main achievement of the study. Unfortunately, this important conclusion is not adequately developed or linked to the existing literature on the prebiotic activity of such compounds.

Response:

To address this observation, this section was rewritten, which can be seen on line 1020.

Round 2

Reviewer 1 Report

Comments and Suggestions for Authors

I am generally satisfied with the authors' responses. 

Some proofreading is still needed. For example:

line 351: Milii used instead of Milli

line 414: large gap

line 573: dash missing in "Triton X 100"

Line 575: SDS should not be pluralized, assuming they are referring to sodium dodecyl sulfate

line 815: "??"

line 998: Ademas is a Spanish word

line 1023: Behnken is misspelled

Author Response

Reviewer 1

Dear reviewer, thank you very much for all your comments. Below are the improvements made to fix it.

 Comment 1) line 351: Milii used instead of Milli

Response:

Thanks for your observation. The text was modified at line 349.

Comment 2) line 414: large gap

Response:

Thanks for your observation. The text was modified at line 411.

Comment 3) line 573: dash missing in "Triton X 100"

Response:

Thanks for your observation. The text was modified at line 568.

Comment 4) Line 575: SDS should not be pluralized, assuming they are referring to sodium dodecyl sulfate

Response:

Thanks for your observation. The text was modified at line 570.

Comment 5) line 815: "??"

Response:

Thanks for your observation. The text was rewritten at line 809.

Comment 6) line 998: Ademas is a Spanish word

Response:

Thanks for your observation. The text was modified at line 997.

Reviewer 2 Report

Comments and Suggestions for Authors

The authors have adressed all my comments properly.

Author Response

We sincerely appreciate the time and effort devoted to reviewing our manuscript. The comments provided by the reviewer have been invaluable in enhancing the quality and clarity of our work. Their observations have allowed us to strengthen the presentation of our results and address key aspects that enrich scientific discussion. We are grateful for their constructive feedback and have carefully incorporated the suggestions into the revised version of the manuscript.

We submitted the revised manuscript, submitting a file with all changes highlighted for clarity. Additionally, at the end of this response letter, we address the questions and comments raised, offering detailed explanations of the changes made to each section. We hope these revisions adequately address the concerns and improve the overall quality of the manuscript.Thank you for your consideration.

Sincerely
